# PhaseFool: Phase-oriented Audio Adversarial Examples via Energy Dissipation

## Abstract

Audio adversarial attacks design perturbations onto inputs that lead an automatic speech recognition (ASR) model to predict incorrect outputs. Current audio adversarial attacks optimize perturbations with different constraints (e.g. $l_p$-norm for waveform or the principle of auditory masking for magnitude spectrogram) to achieve their imperceptibility. However, the existing audio adversarial attacks neglect the influence of phase spectrogram. In this work, we propose a novel phase-oriented algorithm named PhaseFool that can efficiently construct imperceptible audio adversarial examples with energy dissipation. Specifically, we leverage the spectrogram consistency of short-time Fourier transform (STFT) to adversarially transfer phase perturbations to the adjacent frames of magnitude spectrogram and dissipate the energy patterns in the spectrogram. Moreover, we propose a weighted loss function to improve the imperceptibility of PhaseFool. Experimental results demonstrate that PhaseFool can generate full-sentence imperceptible audio adversarial examples and achieve a 6.64x generation speed-up over current state-of-the-art imperceptible counterparts (the imperceptibility is verified through a human study[1]). Further robustness evaluations demonstrate that the adversarial examples constructed by PhaseFool remain effective even after applying realistic simulated environmental distortions and several state-of-the-art defenses. Most importantly, our PhaseFool is the first to exploit the phase-oriented energy dissipation in the audio adversarial example rather than add perturbations on the audio waveform like most of the previous works.

## 1 Introduction

Adversarial examples are specifically designed instances that cause a machine-learning algorithm to produce a misclassification (Szegedy et al., 2013; Biggio et al., 2013). Since researchers have successfully created adversarial examples to make automatic speech recognition (ASR) system transcribe the audio as any different sentence (Carlini & Wagner, 2018; Vaidya et al., 2015; Carlini et al., 2016), audio adversarial attacks have become more and more popular. Those attack algorithms usually optimize perturbations added on the benign audio waveform with different constraints based on $l_p$-norm (Carlini & Wagner, 2018; Zhang et al., 2020). Although the perturbations introduced is small in magnitude, noises in the audio adversarial examples are usually perceptible. Later, algorithms based on psychoacoustic principle (Qin et al., 2019; Schönherr et al., 2019) successfully generate effectively imperceptible adversarial examples by hiding the adversarial perturbations to regions of magnitude spectrogram where it will not be heard by a human.

In current imperceptible targeted audio adversarial attacks, only waveform and magnitude spectrogram are taken into account leaving phase and its nature unexplored. Since the structure of natural speech is highly complicated, these two types of algorithms face several challenges:

- Waveform-oriented attacks are usually not imperceptible enough. Although constraining perturbations with $l_2$-norm or $l_\infty$-norm (Carlini & Wagner, 2018; Liu et al., 2020), these attacks introduce audible noises to the adversarial examples. Later works focused on the temporal dependency (Zhang et al., 2020) successfully reduce human-perceptible noises

---

[1]Generated audio adversarial examples can be found at our demo website: `https://phasefool-demo.github.io/phasefool-demo/`.

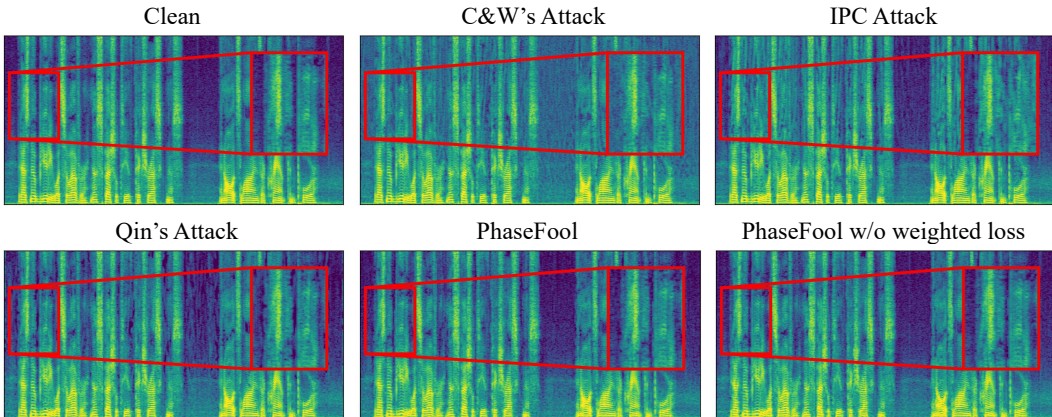

Figure 1: Visualizations of the ground-truth mel-spectrograms and the mel-spectrograms generated by several state-of-the-art attacks and our PhaseFool. Attacks based on the $l_p$-norm (e.g. C&W's attack (Carlini & Wagner, 2018) and IPC attack (Zhang et al., 2020)) introduce obvious perturbations to the mel-spectrogram and cause perceptible noises. Qin's attack leverages the psychoacoustic principle of auditory masking and hides noises in the inaudible regions while our PhaseFool dissipates the energy patterns that are important for classification while less important for human perception.

but their perturbations are still audible, due to the uncorrelated relationship between psychoacoustic perception and $l_p$-norm metrics.

- The optimization of attacks based on the psychoacoustic principles is highly time-consuming and hard to converge. Due to the determining influence of magnitude spectrogram on human perception, the attack that leverages the psychoacoustic principle of auditory masking (Qin et al., 2019) successfully generates imperceptible audio adversarial examples. However, it needs two-stage optimization where the first stage of optimization focuses on finding a relatively small perturbation to fool the network (as is done in prior work (Carlini & Wagner, 2018)) and the second stage makes the adversarial examples imperceptible. The search process of the optimal balance between hiding the adversarial perturbations in the inaudible regions and fooling the ASR systems is empirically difficult. Generating a single full-sentence adversarial example requires approximately 5000 iterations until it finally converges to an imperceptible solution.

This research is motivated by a psychoacoustic principle that phase manipulation in one-channel audios imposes tiny effects on human perception (Vary & Eurasip, 1985). Besides, the inherent spectrogram consistency of short-time Fourier transform (STFT) reveals the fact that the adversarial perturbations in a frame of phase spectrogram will transfer to adjacent frames of magnitude spectrogram (Griffin & Lim, 1984) and dissipate the energy patterns in the spectrogram. These observation motivates us to exploit phase spectrogram for generating audio adversarial examples.

Considering the relationship between phase spectrogram and the spectrogram consistency of STFT, in this work, we propose a novel phase-oriented algorithm called PhaseFool that efficiently generates imperceptible audio adversarial examples via energy dissipation. We prove that the phase-oriented perturbations always reduce the magnitude of the spectrogram in the corresponding areas via spectrogram consistency, which is called energy dissipation. Leveraging such a phenomenon, we bring new directions for audio adversarial attacks and are able to construct imperceptible audio adversarial examples efficiently. To the best of our knowledge, we are the first to consider phase spectrogram in the audio adversarial attacks. Our contributions can be summarized as follows:

- We leverage the spectrogram consistency of STFT to construct phase-oriented audio adversarial examples, which brings new directions for imperceptible audio adversarial attacks.

- We investigate the relationship between phase perturbations and magnitude spectrogram, which is called energy dissipation. Instead of the common $l_p$ distance metrics or magnitude-oriented metrics based on the psychoacoustic principle, we propose a weighted

loss function based on the phenomenon of energy dissipation to improve the imperceptibility of phase-oriented audio adversarial examples.

- We further evaluate the robustness of PhaseFool by constructing adversarial examples that can remain effective even after applying realistic simulated environmental distortions and several state-of-the-art defenses.

We conduct experiments on the LibriSpeech dataset (Panayotov et al., 2015) and implement successful attacks against a pretrained DeepSpeech 2 model (Amodei et al., 2016) and an end-to-end Transformer-based ASR system (Winata et al., 2020). The results show that in terms of imperceptibility, PhaseFool greatly surpasses current state-of-the-art algorithms based on $l_p$-norm metrics and shows comparable imperceptibility with the attacks based on the psychoacoustic principle of auditory masking (verified by a human study). Besides, PhaseFool achieves a 6.64x generation speed-up over current state-of-the-art imperceptible counterparts. Furthermore, in the robustness evaluations, the adversarial examples constructed by PhaseFool remain effective even after applying realistic simulated environmental distortions and several state-of-the-art defenses. We attach the adversarial examples in the experiments and the implementation for reproducibility in the supplementary material.

## 2 BACKGROUNDS

In this section, we briefly overview the background of this work, including audio adversarial examples and phase-aware audio signal processing.

**Audio Adversarial Examples.** After the early works (Gong & Poellabauer, 2017; Cissé et al., 2017) towards neural networks based ASR systems successfully generate untargeted adversarial examples that produce incorrect but arbitrary transcriptions. Recently, more and more efforts have been made in the audio domain (Alzantot et al., 2018; Carlini & Wagner, 2018). Among them, one of the most outstanding studies is the iterative optimization-based attack (Carlini & Wagner, 2018), which introduces a two-stage attack and directly generates targeted adversarial examples using a gradient-descent minimization method. Nevertheless, the generated adversarial examples tend to include noticeable noise. Thus, two recent approaches (Qin et al., 2019; Schönherr et al., 2019) introduce hearing thresholds for designing inaudible perturbations by curbing the signal variation below the threshold of human perception. However, these imperceptible attacks require two stages of optimization and suffer from slow convergence rates.

**Phase-aware Audio Signal Processing.** The interest in utilizing phase information in the field of audio signal processing is constantly increasing (Mowlaee et al., 2016). The STFT phase has been widely used in signal enhancement (Miyahara & Sugiyama, 2014; Choi et al., 2019; Hu et al., 2020) and signal reconstruction (Mowlaee & Kulmer, 2015). The phase in the harmonic domain is used in the speech synthesis (Degottex & Erro, 2014; Ai & Ling, 2020) and the phase in perceptual bands is used in speech watermarking (Wang et al., 2021). The important information in the phase brings positive contributions to the above fields. Nevertheless, the existing audio adversarial attacks directly neglect the influence of phase spectrogram (Schönherr et al., 2019). In this work, instead of introducing perturbations to the audio waveform like most of the previous works, we propose to leverage the spectrogram consistency and generate phase-oriented perturbations for constructing imperceptible audio adversarial examples.

## 3 PHASE-ORIENTED AUDIO ADVERSARIAL EXAMPLES VIA ENERGY DISSIPATION

In this section, we first define the preliminaries of the phase-oriented audio adversarial attack. Then we investigate the relation between phase spectrogram and magnitude spectrogram under spectrogram consistency and prove that phase perturbations always reduce the magnitude of the spectrogram in the corresponding areas under spectrogram consistency, which is called energy dissipation (as is illustrated in Figure 1). Finally, we design a novel phase-oriented loss function based on energy dissipation for constructing imperceptible audio adversarial examples.

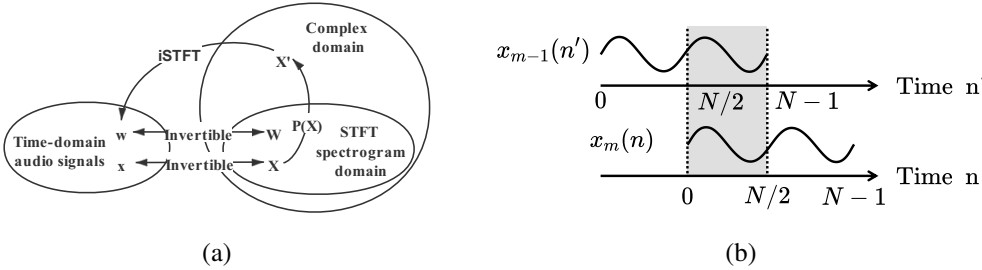

Figure 2: The Illustration of spectrogram consistency. (a) demonstrates that if the spectrogram $X$ is perturbed by $P(X)$, it usually locates outside of the STFT spectrogram domain making it non-invertible. (b) is the overlap-add process with 50% overlapped signals in the inverse STFT.

## 3.1 PRELIMINARIES OF PHASE-ORIENTED AUDIO ADVERSARIAL ATTACK

### 3.1.1 PROBLEM DEFINITION

Given an input audio waveform $x$ and a transcription $y$ corresponding to $x$. Denote the automatic speech recognition (ASR) system as $f(\cdot)$ that outputs a transcription. The objective of the targeted audio adversarial attack is to find a small perturbation $\delta$ that constructs an imperceptible and targeted example $x' = x + \delta$ to fool the ASR system $f(\cdot)$ so that $f(x') = y_t$, where $y_t$ is the targeted adversarial transcription. Denote $\mathcal{L}_{net}(f(x'), y_t)$ as the loss function for the ASR model that is minimized when $f(x') = y_t$. Then the general formulation of targeted adversarial attack for ASR system can be defined as:

$$\underset{\delta}{\text{argmin}} \quad \mathcal{L}_{net}(f(x + \delta), y_t) + \lambda \|\delta\|_p^p , \text{ s.t.} \|\delta\|_\infty < \epsilon , \tag{1}$$

where $\lambda > 0$ is the trade-off parameter between the imperceptibility and loss function $\mathcal{L}_{net}(f(x + \delta), y_t)$ (in some formulations $\lambda = 0$), $\epsilon$ control the maximum perturbation value, and p could be specified as different values, according to the attacker's requirement (e.g. $p = 2$ in C&W's attack (Carlini & Wagner, 2018)). Then we will describe how to generate phase-oriented and targeted audio adversarial examples in the next subsection.

### 3.1.2 PHASE-ORIENTED ADVERSARIAL EXAMPLE GENERATION

Denote $\phi(\cdot)$ as the function of the STFT and $\phi^{-1}(\cdot)$ as the function of the inverse STFT and $X = \phi(x) = |X| \cdot e^{i\theta_x}$, where $|X|$ and $\theta_x = \angle X$ representing the spectral magnitude and the instantaneous phase spectrogram, respectively. Denote $\mathcal{T}(X) = \phi(\phi^{-1}(X))$. We assume that window function $w$, window length $N$ and window shift $R$ is fixed and a signal can be perfectly reconstructed from its STFT spectrogram through the inverse STFT in this setting. We propose to generate perturbations on the instantaneous phase spectrogram and reconstruct the audio adversarial examples with the inverse STFT as the following formulation:

$$\underset{\delta}{\text{argmin}} \quad \mathcal{L}_{net}(f(\mathcal{E}(\phi^{-1}(|X| \cdot e^{i\theta_x} \cdot e^{j\delta}))), y_t), \text{ s.t.} -\pi \le \delta \le \pi , \tag{2}$$

where $\mathcal{E}(\cdot)$ is the feature extraction function for the ASR model and the $\phi^{-1}(|X| \cdot e^{i\theta_x} \cdot e^{j\delta})$ is the generated audio adversarial example.

As shown in Figure 2, the spectrogram $X' = |X| \cdot e^{i\theta_x} \cdot e^{j\delta}$ perturbed by $P(X)$ is usually located outside the STFT spectrogram domain. Considering the phenomenon of spectrogram consistency (Le Roux et al., 2010), it is easy to prove the following two conclusions: 1) $|X'| \ne |\mathcal{T}(X')|$ mathematically; 2) $\phi^{-1}(X')$ and $\phi^{-1}(\mathcal{T}(X'))$ represent the same time-domain audio signals. Since the STFT and iSTFT functions are differentiable, these conclusions enable us to adversarially manipulate the reconstructed magnitude spectrogram by introducing perturbations to the original phase spectrogram, which is called phase-oriented audio adversarial attack. Then we detail how the perturbations on a frame of phase spectrogram influence the adjacent frames of corresponding magnitude spectrogram and reveal the energy dissipation nature of the phase-oriented audio adversarial attack.

## 3.2 RELATION ANALYSIS BETWEEN PHASE AND MAGNITUDE

This subsection mainly discusses the essence of the relationship between phase perturbations and magnitude spectrogram frames. Reconstructing the time-domain signals after introducing perturbations to the phase spectrogram includes an overlap-add process. For simplicity, we assume that the window shift length $R$ is equal to half of the window length $N$, which means the adjacent frames have 50% overlapped signals. As shown in Figure 2, the signals $x_{m-1}(n')$ and $x_m(n)$ in adjacent frames $m-1$ and $m$ overlap, where $n'$ and $n$ represent the time indices and $n' = n + \frac{N}{2}$. $x_m(n)$ are obtained by an inverse Fourier transform as the following:

$$x_m(n) = \phi^{-1}(X_m) = \frac{1}{N} \sum_{k=0}^{N-1} X_m[k] e^{j\frac{2\pi kn}{N}} , \tag{3}$$

where $j$ is the imaginary unit and $X_m[k]$ are spectrogram samples in frames $m$ at $k$-th frequency bin. $x_{m-1}(n')$ can also be obtained with Equation 3. Then we introduce perturbations $\delta_m$ to the frame $m$ and the corresponding signal $\hat{x}_m(n)$ becomes:

$$\hat{x}_m(n) = \frac{1}{N} \sum_{k=0}^{N-1} X_m[k] e^{j\frac{2\pi kn}{N}} e^{j\delta_m[k]} . \tag{4}$$

Since the human auditory system is relatively less sensitive to the tiny time delay caused by phase rotation, the phase perturbations are inherently more imperceptible than the perturbations on the audio waveform. Then we further investigate how phase perturbations affect the magnitude spectrogram. The reconstructed signal $\tilde{x}_m(n)$ with the overlap-add process is obtained as:

$$\tilde{x}_m(n) = \hat{x}_{m-1}(n') + \hat{x}_m(n) = \frac{1}{N} \sum_{k=0}^{N-1} X_{m-1}[k] e^{j\frac{2\pi kn'}{N}} e^{j\delta_{m-1}[k]} + \frac{1}{N} \sum_{k=0}^{N-1} X_m[k] e^{j\frac{2\pi kn}{N}} e^{j\delta_m[k]} . \tag{5}$$

We substitute $X_{m-1}[k]$ and $X_m[k]$ with the Fourier transform of $x_{m-1}(n')$ and $x_m(n)$, then the overlapping area of $\tilde{x}_m(n)$ in Equation 5 will become:

$$\overline{x}_m(n) = \frac{1}{N} \sum_{k=0}^{N-1} \left( \sum_{n'=N/2}^{N-1} x_{m-1}(n') e^{j\delta_{m-1}[k]} + \sum_{n=0}^{N/2-1} x_m(n) e^{j\phi_m[k]} \right) . \tag{6}$$

The overlapping area of the $x_{m-1}(n')$ and $x_m(n)$ in Figure 2 (b) satisfies the following relationship based on spectrogram consistency:

$$x_{m-1}(t + \frac{N}{2}) = x_m(t), \ s.t. \ 0 \leq t \leq \frac{N}{2} . \tag{7}$$

With Equation 7, we can deduce the final formulation for the reconstructed signal $\overline{x}_m(n)$ as following:

$$\overline{x}_m(n) = \frac{1}{N} \sum_{n=0}^{N/2-1} x_m(n) \sum_{k=0}^{N-1} \left( e^{j\delta_{m-1}[k]} + e^{j\delta_m[k]} \right) , \tag{8}$$

where the clean audio can be represented by setting $\delta_{m-1}[k] = 0$ and $\delta_m[k] = 0$. Therefore, the ratio $\gamma_m[k]$ of the reconstructed signal's spectrogram energy to the clean audio's satisfies:

$$\gamma_m[k] = \frac{\left| e^{j\delta_{m-1}[k]} + e^{j\delta_m[k]} \right|}{\left| e^{j0} + e^{j0} \right|} = \frac{\sqrt{2 + 2\cos(\delta_{m-1}[k] - \delta_m[k])}}{2} , \tag{9}$$

where $\gamma_m[k]$ is ranged within $[0, 1]$. It means that the phase perturbations $\delta_{m-1}[k], \delta_m[k]$ always cause loss of the reconstructed magnitude spectrogram, which is called 'energy dissipation'. Till now, we reveal the essence of phase-oriented audio adversarial attack: it slightly dissipates the energy patterns in the spectrogram and constructs the adversarial examples with the loss function in Equation 2 in the meantime.

### 3.3 OPTIMIZATION WITH ENERGY DISSIPATION

Simply minimizing the $l_p$ distortion of $(1 - \gamma)$ is a possible solution. However, the $l_p$ based metrics yield more audible perturbations in the practical attacks due to the differences between $l_p$ distance and human perceptibility. In this paper, we propose a weighted loss function to constrain phase perturbations in an imperceptible range. Since the perturbations of our attack always cause energy loss to the spectrogram, this function should dissipate the energy that is crucial for the classification model and meanwhile preserve the energy that is important for human perception. According to the characteristics of human auditory systems, we divide the spectrogram into three parts: 1) the signals that are important for human perception; 2) the signals that are relatively unimportant for human auditory systems; 3) the inaudible signals that are below the threshold in quiet; For 1) and 2), we manage to constrain their $\gamma$ in different levels $\beta_1$ and $\beta_2$, respectively. For 3), we do not constrain the corresponding $\gamma$ since the energy dissipation in these areas can not be perceived. We divide quiet signals 3) following Lin & Abdulla (2015). The signals 1) are the local maxima that have the largest magnitude within 3 bins in the spectrum frames (in order to preserve the signals in high frequency we do not adopt Bark scale) and the signals 2) are the rest of the signals in the spectrogram.

**Weighted Loss function.** We formulate the construction of imperceptible phase-oriented adversarial examples as minimizing the following weighted loss function:

$$\mathcal{L} = \mathcal{L}_{\text{net}}(f(\mathcal{E}(\phi^{-1}(|X| \cdot e^{i\theta_x} \cdot e^{j\delta}))), y_t) + \lambda \cdot \mathcal{L}_{\text{imp}}(\gamma) , \tag{10}$$

$$\mathcal{L}_{\text{imp}}(\gamma) = \alpha_1 \cdot \sum_{m=0}^{N-1} \sum_{k=0}^{N-1} \mathcal{I}_1(max\{\beta_1 - \gamma_m[k], 0\}) + \alpha_2 \cdot \sum_{m=0}^{N-1} \sum_{k=0}^{N-1} \mathcal{I}_2(max\{\beta_2 - \gamma_m[k], 0\}) , \tag{11}$$

where $\mathcal{I}_1$ and $\mathcal{I}_2$ are the indicator functions that are equal to 1 when the $\gamma_m[k]$ is in the corresponding part in the spectrogram and $\alpha_1 = 1$, $\alpha_2 = 0.05$ are trade-off parameters. We initialize the parameter $\lambda$ as $1 \times 10^{-4}$ and update $\lambda$ every 10 iterations according to the performance of the attack. Since the hinge loss is utilized to constrains the dissipation ratio $\gamma$ of the part 1) and 2) above the levels $\beta_1 = 0.98$ and $\beta_2 = 0.90$, the tiny energy losses caused by the phase perturbations will be perceived as almost inaudible volume differences instead of the echoes or noises;

## 4 EXPERIMENTAL SETUP

**Datasets.** We conduct experiments on the LibriSpeech dataset (Panayotov et al., 2015), which contains 1000 hours of 16KHz English speech derived from audiobooks. We randomly select 100 audio examples from the test-clean dataset as source examples, and 100 separate transcriptions from the same dataset to be the targeted transcriptions following the experimental setup with (Carlini & Wagner, 2018; Qin et al., 2019). We ensure that each target transcription is around the same length as the original transcription because it is unrealistic to perturb a short audio sample (e.g., 10 words) to have a much longer transcription (e.g., 30 words). Examples of the original and targeted transcriptions are available in the supplementary material.

**Implementation Details.** We successfully implement our PhaseFool and several state-of-the-art attack baselines against a pretrained DeepSpeech 2 model (Amodei et al., 2016) and an end-to-end speech recognition model based on the low-rank Transformer architecture following Winata et al. (2020).[2] The DeepSpeech 2 model is an open-source ASR system with Connectionist Temporal Classification (CTC) method (Graves et al., 2006) and LSTM as its main components, which takes the log spectrograms as the input features. And the low-rank Transformer-based ASR system that takes log-mel spectrograms as its input features is composed of lightweight transformer architecture and is trained with the cross-entropy loss. Since the feature extraction is differentiable, our phase attack can be easily extended to other ASR systems. All experiments are carried out on 1 Nvidia GTX 3080 GPU with 10 GB memory. We use Adam (Kingma & Ba, 2015) as the optimizer with $\beta_1 = 0.9$, $\beta_2 = 0.98$, $\epsilon = 10^{-9}$ and the learning rate is set as $5 \times 10^{-3}$.

---

[2]The DeepSpeech 2 model follows the Pytorch implementation in the adversarial robustness toolbox (Nicolae et al., 2018) and the low-rank Transformer model is implemented following Winata et al. (2020).

**Threat Model.** As is done in most previous works, we assume a white-box threat model, i.e. the adversary is allowed to compute gradients through the model in order to generate adversarial examples. Considering that current white-box audio adversarial attacks (Carlini & Wagner, 2018; Yakura & Sakuma, 2019; Qin et al., 2019) have been widely applied into the black-box attacks (Yuan et al., 2018; Chen et al., 2020) as the base generation methods and shown great improvements in terms of robustness or imperceptibility, our work only focus on investigating the white-box model, as is done in most previous works.

**Evaluation Metrics.** 1) Word error rate (WER). We calculate the WER metric which is defined as $WER = \frac{S+D+I}{N} \times 100\%$, where $S$, $D$, and $I$ are the numbers of substitutions, deletions, and insertions, respectively, and N is the total number of words. Particularly, we choose the targeted transcription as the label for calculating the WER in the experiments[3].; 2) Success Rate (SR). SR is the ratio of examples that can be successfully recognized as the malicious target texts by ASR systems. Here, "successfully recognized" means that the adversarial example is recognized as the malicious target transcription;

## 5 RESULTS

In this section, we evaluate the performance of PhaseFool in terms of imperceptibility, generation speedup, robustness against simulated environmental distortions and state-of-the-art defenses. The extra analysis and ablation studies can be found in Appendix A.

### 5.1 IMPERCEPTIBILITY ANALYSIS

**Experimental Design.** We construct 100 full-sentence targeted adversarial examples sampled from the LibriSpeech test-clean dataset for our PhaseFool attack. For comparison, we then generate adversarial examples using the prior state-of-the-art works as baselines, including: 1) *C&W's attack (Carlini & Wagner, 2018)*, a two-stage adversarial attack for audio based on CTC loss and $l_p$-norm; 2) *IPC attack (Zhang et al., 2020)*, a new Iterative Proportional Clipping (IPC) algorithm that generates robust audio adversarial examples with temporal dependency; 3) *Qin's attack (Qin et al., 2019)*, the imperceptible attack that leverages the psychoacoustic principle of auditory masking. We ensure that all attacks have a 100% success rate and the target transcription is consistent among the experiments to exclude other interference factors. The user study is conducted to verify the imperceptibility of these attacks. We recruit 24 users online and ask them to tell us if the examples have any background noise (e.g., static, echoing, people talking in the background) following (Qin et al., 2019), which is the most representative experiment of how an attack would work in practice. The users are asked to listen to each sample with headphones on, and answer questions about a set of audio samples. In all experiments, users can replay the audio samples multiple times.

**Result Analysis.** As shown in Table 1, 31% of the users believe that the clean audio samples contain noises. Only 43% and 46% of the users believe that the adversarial examples generated by the Phase-Fool against the DeepSpeech 2 system and low-rank Transformer system contain some noises, which is similar to Qin's attack. Meanwhile, PhaseFool and Qin's attack greatly surpass the $l_p$-norm based C&W's attack and IPC attack. It can be seen that our phase-oriented attack shows comparable imperceptibility with the attacks that leverage the psychoacoustic principle of auditory masking. However, our attack needs fewer steps and only one optimization stage to generate imperceptible audio adversarial examples rather than complex two optimizations stages in Qin et al. (2019). We will further explore the generation speed of these attacks in the next subsection.

| Method | Classified as Noisy ↓ | |
| --- | --- | --- |
| | DeepSpeech 2 | Transformer |
| Clean | 31% | 31% |
| C&W's Attack | 75% | 77% |
| IPC Attack | 59% | 68% |
| Qin's Attack | 49% | **45%** |
| PhaseFool | **43%** | 46% |

Table 1: Results of the user study for imperceptibility. '↓' means lower is more imperceptible.

---

[3]This calculation method of WER follows Carlini & Wagner (2018); Qin et al. (2019); Liu et al. (2020)

| Method | DeepSpeech 2 | | Transformer | |
|---|---|---|---|---|
| | Steps ↓ | Convergence Times (s) ↓ | Steps ↓ | Convergence Times (s) ↓ |
| C&W's Attack | 4000 | $1031.16 \pm 369.03$ | 4000 | $211.87 \pm 12.31$ |
| IPC Attack | 3000 | $871.72 \pm 328.35$ | 3000 | $158.75 \pm 10.99$ |
| Qin's Attack | 5000 | $1310.86 \pm 583.28$ | 5000 | $305.91 \pm 8.02$ |
| PhaseFool | **700** | **197.85** $\pm$ **56.09** | **500** | **33.10** $\pm$ **3.27** |

Table 2: Comparison of the generation speed with 95% confidence intervals. Convergence Times (s) means that the algorithm can generate a successful audio adversarial examples within the listed seconds on average.

| Method | Defenses | | | |
|---|---|---|---|---|
| | TD | Dropout | Downsample | Noise |
| C&W's Attack | 57.78% | 63.75% | 67.05% | 63.70% |
| IPC Attack | 52.67% | 71.62% | 68.95% | 74.42% |
| Qin's Attack | 58.22% | 61.20% | 66.74% | 82.64% |
| PhaseFool | 57.02% | 64.09% | 68.60% | 62.81% |

Table 3: Area under curve (AUC) scores for different defenses against PhaseFool and various baseline attacks.

## 5.2 GENERATION SPEEDUP

We evaluate the generation speed of PhaseFool compared with several state-of-the-art audio adversarial attacks in the previous experiment. We generate 100 adversarial examples for each algorithm with the 100% success rate and ensure that the original audio samples and the target transcriptions are consistent among the experiments to exclude other interference factors. The lengths of the selected audio range from 3 seconds to 10 seconds (5.18 seconds on average) and the detailed analysis of the selected samples can be found in the Appendix A.4. As shown in Table 2, it can be seen that PhaseFool can generate a successful audio adversarial example for the low-rank Transformer system in around 30 seconds on average, which is a 9.24x speedup compared with the imperceptible counterpart Qin's attack (Qin et al., 2019). Moreover, for the DeepSpeech 2 system, Qin's attack usually requires over 20 minutes to generate an adversarial example while our PhaseFool only needs around 3 minutes. Although the constraints used in the baseline attacks can improve their imperceptibility, these constraints significantly slow down the convergence rates of the optimization process. Besides, the two-stage optimizations make C&W's attack and Qin's attack requires even more steps to generate adversarial examples. Empirically, the computational complexity of audio adversarial attack mainly depends on iteration steps and the attacked model $f(\cdot)$. Since our PhaseFool is a one-stage attack without complex computations and only requires 700 steps to obtain a successful adversarial example, the generation speed of PhaseFool is much faster than the baseline attacks.

## 5.3 ROBUSTNESS ANALYSIS

We conduct the robustness analysis to explore whether the adversarial examples constructed by PhaseFool can remain effective even after applying realistic simulated environmental distortions and several state-of-the-art defenses. We generate 100 adversarial examples only for the DeepSpeech 2 system in this experiment. We adopt the Expectation over Transformation (EOT) algorithm (Athalye et al., 2018) to simulate transformations caused by environmental distortions and defenses and incorporate the transformations into the generation process following the previous works (Yakura & Sakuma, 2019; Qin et al., 2019). The details of the experimental setup can be found in Appendix A.1. The results are shown in Table 3 and Table 4.

**Robustness Against Defenses.** To compare the robustness of attacks against adversarial audio defenses, PhaseFool and the baseline attacks are applied with the following defenses: 1) *Temporal Dependency (TD) defense (Yang et al., 2019)*, the defense that uses the temporal dependency in audio data to gain discriminate power against audio adversarial examples; 2) *Dropout*

| Method | Reverberation | | | Noise | | |
|---|---|---|---|---|---|---|
| | SR ↑ | WER ↓ | Classified as Noisy ↓ | SR ↑ | WER ↓ | Classified as Noisy ↓ |
| C&W's Attack | 47% | 22.69% | 100% | 55% | 15.38% | 88% |
| IPC Attack | 21% | 37.07% | 97% | 37% | 27.94% | 73% |
| Qin's Attack | 43% | 26.44% | 85% | 19% | 43.45% | 51% |
| PhaseFool | 38% | 29.62% | 61% | 44% | 16.34% | 48% |

Table 4: The results of robustness evaluation against reverberant and noisy environments.

*defense (Jayashankar et al., 2020)*, the defense based on the uncertainty introduced by dropout in neural networks; 3) *Downsample defense (Yang et al., 2019)*, a common defense that down-samples a band-limited audio file without sacrificing the quality of the recovered signal while mitigating the adversarial perturbations in the reconstruction phase. 4) *Noise defense*, a common defense that introduces white Gaussian noise to disable the adversarial perturbations. We follow the same experimental procedures as the corresponding defenses and adopt their evaluation metric: the area under curve (AUC) score. As shown in Table 3, the AUC scores for PhaseFool's adversarial examples against the four defenses are mostly distributed between 57%∼68%, which means that the classifiers of these defenses have poor performance on our adversarial examples. However, although IPC attack shows great performance against the TD defense, its adversarial examples are easily detected by the dropout defense and the noise defense. Besides, Qin's attack is also not robust enough with the noise defense. These results demonstrate the robustness of our phase-oriented adversarial examples against several sophisticated defenses.

**Robustness Against Environmental Distortions.** To demonstrate the robustness of PhaseFool against environmental distortions, we evaluate the attack success rate (SR) and the word error rate (WER) of adversarial examples generated by PhaseFool and baseline attacks in the simulated environments that contain reverberations and white Gaussian noise following the previous works (Qin et al., 2019; Yakura & Sakuma, 2019). Besides, to investigate whether the adversarial examples of PhaseFool can still remain imperceptible to the human ear in these scenarios, we also conduct the user study with the same experimental settings in Subsection 5.1. As shown in Table 4, the adversarial examples generated by PhaseFool achieve approximately 40% attack success rate in the reverberant and noisy environments. However, the imperceptibility of Qin's attack significantly drops in the reverberation environment, while the imperceptibility of our PhaseFool does not degrade so much relatively. Since the phase-oriented adversarial examples sound more like echoes or blurred pronunciation, our PhaseFool shows better imperceptibility in the attacks for the environments that contain distortions.

## 6 Conclusions

In this work, we propose PhaseFool, a phase-oriented audio adversarial attack, which leverages energy dissipation to efficiently construct imperceptible audio adversarial examples. Leveraging the phenomenon of energy dissipation, we bring new directions for audio adversarial attacks. Moreover, we propose a weighted loss function to improve the imperceptibility of phase-oriented audio adversarial examples. Experiments on the LibriSpeech dataset (Panayotov et al., 2015) demonstrate that our PhaseFool significantly surpasses current state-of-the-art algorithms based on $l_p$-norm metrics and shows comparable imperceptibility with the attacks based on the psychoacoustic principle of auditory masking. Besides, PhaseFool achieves a 6.64x generation speed-up over current state-of-the-art imperceptible counterparts. Moreover, in the robustness evaluations, the adversarial examples constructed by PhaseFool remain effective even after applying realistic simulated environmental distortions and several state-of-the-art defenses.

For future work, we will continue to improve the imperceptibility of the phase-oriented audio adversarial examples, and apply PhaseFool to black-box settings for practical attacks at commercial ASR systems. We will also investigate PhaseFool's capacity in online learning scenarios.

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

# A APPENDIX

## A.1 THE DETAILED SETUP OF ROBUSTNESS EVALUATION

In the robustness evaluation, we adopt the Expectation over Transformation (EOT) algorithm (Athalye et al., 2018) to simulate transformations caused by environmental distortions and defenses and incorporate the transformations into the generation process. Then we transcribe the newly obtained audio adversarial examples and finally calculate SR and WER to evaluate their robustness.

### A.1.1 THE SETUP OF EVALUATION AGAINST DEFENSES

For TD defense, we randomly select the first $k$ portion of the adversarial examples ($k$ is chosen from $\{\frac{1}{3}, \frac{1}{2}, \frac{2}{3}, \frac{1}{1}\}$) and adopt the EOT algorithm to make the generated adversarial examples adaptive to

the TD defense. In the defense stage, we choose $k = \frac{1}{2}$ and calculate the AUC scores of WER following (Yang et al., 2019).

For Dropout defense, we randomly choose the dropout rate $p$ of ASR systems from $\{0.00, 0.05, 0.10, 0.15\}$ and adopt the EOT algorithm to make the generated adversarial examples adaptive to the Dropout defense. In the defense stage, we set the defense dropout rate as $p = 0.05$ and calculate the AUC scores of the SVM-F classifier following (Jayashankar et al., 2020).

For Downsample defense, we conduct the EOT algorithm by performing the attack on the down-sampled elements of the original audio sequence. In the attack stage, we randomly choose whether to downsample the adversarial examples from 16000 Hz to 8000 Hz. In the defense stage, we down-sample the audio from 16000 Hz to 8000 Hz and calculate the AUC scores of WER following (Yang et al., 2019).

For Noise defense, we randomly add White Gaussian noise with the boundary of $\pm\triangle$ to the generation process of the adversarial examples. Since the input audio is normalized, here we randomly choose the $\triangle$ from $\{0.01, 0.02, 0.03\}$ in the attack stage. In the defense stage, we choose $\triangle = 0.02$ and calculate the AUC scores of WER.

### A.1.2  THE SETUP OF EVALUATION AGAINST ENVIRONMENTAL DISTORTIONS

For the reverberant scenarios, we use an acoustic room simulator that applies the classic Image Source Method introduced in (Scheibler et al., 2018) to create the room impulse response $r$. We set the room configurations (room dimension, source audio and target microphone's locations, and reverberation time) following Qin et al. (2019). Then, the generated room impulse response $r$ is convolved with the clean audio to create the speech with reverberation. In the attack stage, we generate 1000 random room configurations sampled from the distribution as the training room set. And the test room set includes another 100 random room configurations sampled from the same distribution for the 100 audio adversarial examples generated by different attacks.

For the noisy scenarios, we randomly add White Gaussian noise with the boundary of $\pm\triangle$ to the generation process of the adversarial examples. Since the input audio is normalized, here we randomly choose the $\triangle$ from $\{0.01, 0.02, 0.03\}$. In the test stage, we choose $\triangle = 0.02$. Note that this experiment aims at evaluating the attack success rate and imperceptibility of attacks in the noisy scenarios, which is different from the evaluation against noise defense.

### A.2  ABLATION STUDIES

We evaluate the generation speed and imperceptibility in the ablation studies to explore the influences of the weighted loss function proposed in Equation 11. As shown in Table 5 and Table 6, when we do not constrain the phase perturbations and optimize the example until it is successful, PhaseFool only needs 305 and 212 steps on average to construct an audio adversarial example for the DeepSpeech 2 system and low-rank Transformer system, correspondingly. It can be seen that the examples without constraints are even more imperceptible than the $l_p$-norm based attacks that need thousands of steps. But these examples introduce more noise than PhaseFool with the weighted hinge loss in Equation 11, which demonstrates the effectiveness of our proposed loss function. However, when we constrain the phase perturbations with $l_2$ loss, the noises in adversarial examples are even louder than the examples generated without constraint. It demonstrates the importance of curbing the energy dissipation ratio $\gamma$ above the weighted level for different types of regions with the loss function proposed in Subsection 3.3.

| Method | DeepSpeech 2 | | |
|---|---|---|---|
| | Steps ↓ | Convergence Times (s) ↓ | Classified as Noisy ↓ |
| PhaseFool | 700 | 197.85±56.09 | 43% |
| PhaseFool with $l_2$ loss | 700 | 195.48±57.29 | 75% |
| PhaseFool w/o constraint | 305 | 88.62±233.71 | 64% |

Table 5: The results of ablation studies against the DeepSpeech 2 system.

| Method | Transformer | | |
|---|---|---|---|
| | Steps ↓ | Convergence Times (s) ↓ | Classified as Noisy ↓ |
| PhaseFool | 500 | 33.10±3.27 | 44% |
| PhaseFool with $l_2$ loss | 500 | 32.54±3.50 | 72% |
| PhaseFool w/o constraint | 212 | 17.80±19.56 | 67% |

Table 6: The results of ablation studies against the low-rank Transformer system.

## A.3 INTERPRETATION OF ATTACKS

For further investigating the interpretation of different attacks, we present the pixel-wise difference of the adversarial examples in the mel-spectrogram domain to visualize the learned adversarial perturbations in Figure 3. The perturbations of C&W's attack are disorderly distributed in the mel-spectrogram while Qin s attack and IPC attack tend to hide perturbations in the utterance regions. Our PhaseFool attack provides additional insights about the vulnerability of ASR models, i.e. the influences of different regions in the spectrogram for the audio adversarial attack. For example, as shown in Figure 3, the perturbations generated by our PhaseFool attack mainly occur on the positions around the harmonics in the mel-spectrogram. The perturbation positions reveal the fragile part of the spectrogram for the ASR system's prediction. With the $l_2$ loss function, PhaseFool significantly perturbs the mel-spectrogram regions that are important for human perception and cause audible echoes in the audio adversarial examples. When we remove the constraints of PhaseFool, the magnitude of perturbations in the mel-spectrogram grows obviously, which demonstrates the effectiveness of the proposed weighted loss.

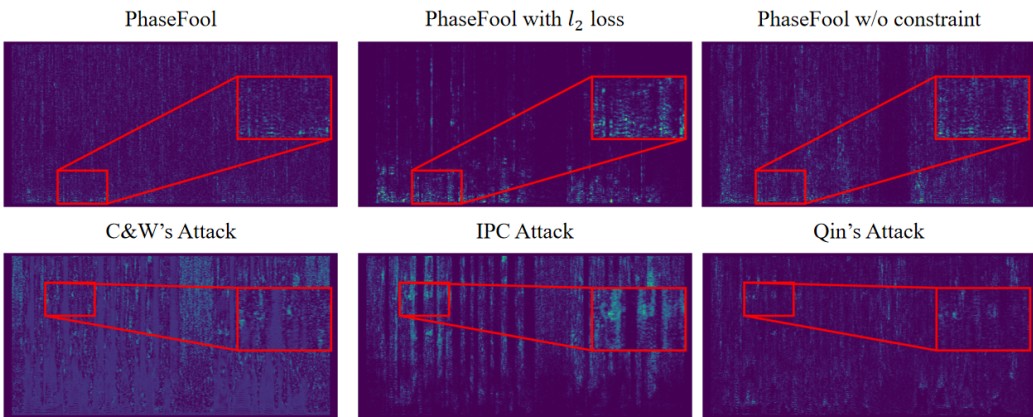

Figure 3: Pixel-wise difference of the generated audio adversarial examples in the mel-spectrogram domain. The perturbations of C&W's attack are disorderly distributed in the mel-spectrogram. Qin's attack and IPC attack tend to hide perturbations in the utterance regions while our PhaseFool attack generates perturbations around the harmonics in the spectrogram.

## A.4 LENGTH ANALYSIS

In all experiments, we randomly select 100 audio examples from the test-clean set of the LibriSpeech dataset (Panayotov et al., 2015) as source examples, and 100 separate transcriptions from the same dataset to be the target transcriptions following the experimental setup with (Carlini & Wagner, 2018; Qin et al., 2019). We ensure that each target transcription is around the same length as the original transcription because it is unrealistic to perturb a short audio sample (e.g., 10 words) to have a much longer transcription (e.g., 30 words). Figures 4 illustrates the comparison of the text length between original transcriptions and target transcriptions. And Figures 5 shows the lengths of 100 selected audio samples in the experiments.

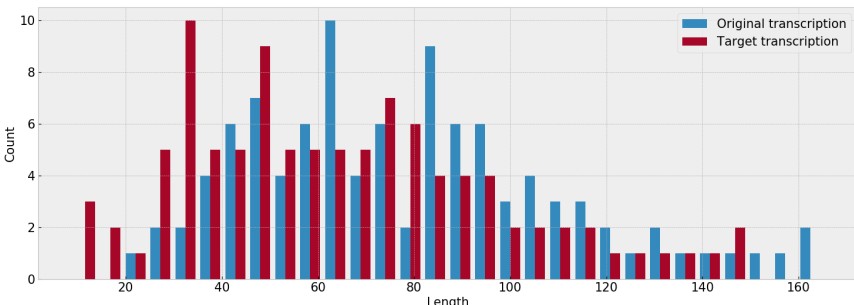

Figure 4: The length distribution of 100 text examples randomly sampled from Librispeech in our experiments.

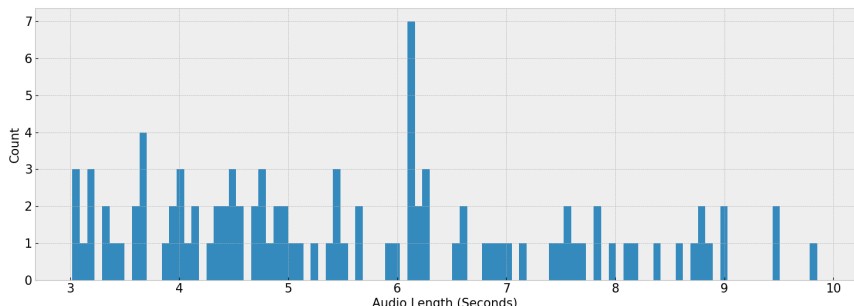

Figure 5: The length distribution of 100 audio examples randomly sampled from Librispeech in our experiments.

## A.5 ANALYSIS ON THE CONVERGENCE OF LOSS AND LEVENSHTEIN DISTANCE

Here we analyze the CTC loss and Levenshtein distance convergence processes when we attack the DeepSpeech 2 model with PhaseFool. We randomly select 20 samples in the experiments. As shown in Figure 6, the CTC loss of most cases quickly converges to a value close to 0. The Levenshtein distance in Figure 6 converges to 0 on all cases before 700 iteration steps. Note that when the value of Levenshtein distance becomes 0, the adversarial example is successfully generated.

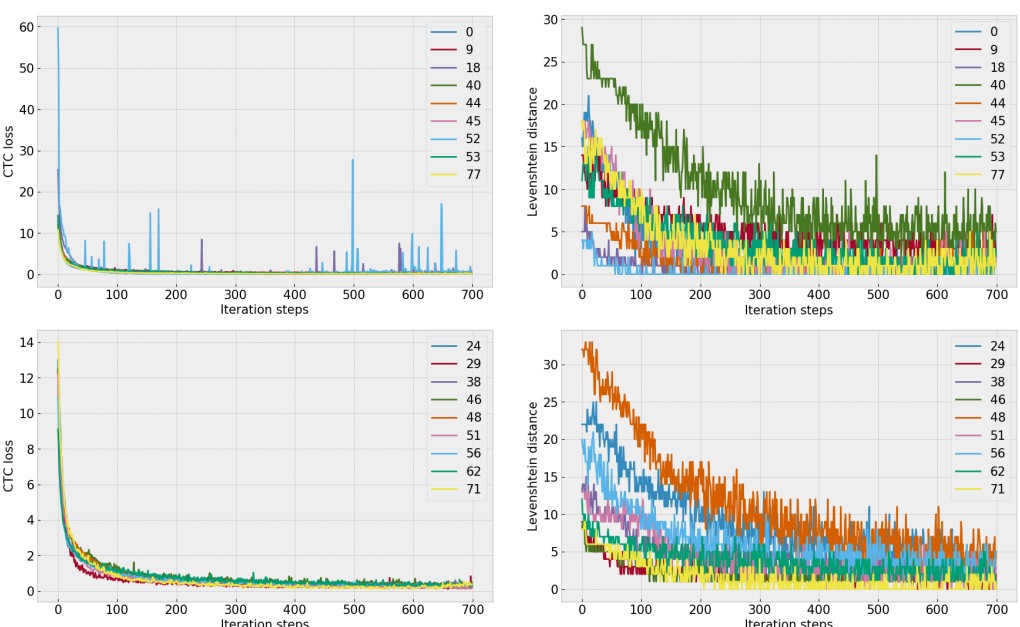

Figure 6: The Illustration of the CTC loss and Levenshtein distance convergence processes of Phase-Fool. We randomly picked 20 samples in the experiments.

