# OpenReview forum: "PhaseFool: Phase-oriented Audio Adversarial Examples via Energy Dissipation"
_ICLR.cc/2022/Conference — ICLR 2022 Submitted_

### Official Review · Reviewer_vUjU · 2021-11-01

**Correctness:** 4
**Technical Novelty And Significance:** 4
**Empirical Novelty And Significance:** 4
**Recommendation:** 8
**Confidence:** 4

**Details Of Ethics Concerns:**

The proposed method can be used to harm neural network-based security systems.


**Main Review:**

Strengths:
1. The idea of phase-based adversarial attack makes sense in that the phase information is usually known as less perceptible than magnitude information.
2. The authors theoretically show that the phase-based perturbation leads to energy dissipation.
3. The experiment results are good, and is even better than the magnitude-based method.

Weakness:
1. It could have been nicer if the authors had experimented with the raw-waveform-based ASR networks.
2. It could have been nicer if the authors had experimented the method on more diverse audio NNs.

**Summary Of The Paper:**

This paper proposes a novel adversarial attack method that attacks ASR network.
The idea is that the attack is likely to be imperceptible when phase information is perturbed because phase information is known to give releatively small perceptible difference than magnitude information.
The perturbation on phase information influences the magnitude information which is known as energy dissipation.
Experiment results show that it reaches almost state-of-the-art results in terms of imperceptibility. The advantage of the proposed method is that it can generate the imperceptible perturbation using fewer steps than the previous state-of-the-art method.
They also qualitatively show that it is usually the harmonic parts of speech where the energy dissipates.


**Summary Of The Review:**

The idea is solid.
Experiment results are good.
Writing is straightforward and easy to follow.
All in all, I recommend to accept the paper.

---

> ### Author Response · Authors · 2021-11-18
> **Response to Reviewer vUjU**
>
> ### [About the evaluation on more diverse audio NNs]
>
> We conduct the evaluations for the pretrained DeepSpeech 2 [1] ASR system with CTC loss. The results are shown in the following tables. We also evaluate our attack against some defenses and simulated environmental distortions. We have added these evaluations in Subsection 5.3 in the rebuttal version of the paper.
>
> *[Annotation] You can refer to Subsection 5.1 and Subsection 5.2 in our rebuttal version of the paper to see the detailed information.*
>
> | Method       | DeepSpeech 2 （Noisy or not） | Transformer（Noisy or not） |
> | ------------ | ----------------------------- | --------------------------- |
> | Clean        | 31%                           | 31%                         |
> | C&W's Attack | 75%                           | 77%                         |
> | IPC Attack   | 59%                           | 68%                         |
> | Qin's Attack | 49%                           | 45%                         |
> | PhaseFool    | 43%                           | 46%                         |
>
> | Method       | DeepSpeech 2 （Convergence Times (seconds)） | Transformer（Convergence Times (seconds)） |
> | ------------ | -------------------------------------------- | ------------------------------------------ |
> | C&W's Attack | 1031.16                                      | 211.87                                     |
> | IPC Attack   | 871.72                                       | 158.75                                     |
> | Qin's Attack | 1310.86                                      | 305.91                                     |
> | PhaseFool    | 197.85                                       | 33.10                                      |
>
>
>
> ## [About the evaluation with the raw-waveform-based ASR networks]
>
> Due to the time limitation of the rebuttal period, we haven't evaluated our PhaseFool against the raw-waveform-based systems. So this will be regarded as our future work. Thanks for your advice!
>
>
>
> [*] We also include other details and explanations of the experiments in the Appendix.
>
>
>
> [1] Amodei, Dario, et al. "Deep speech 2: End-to-end speech recognition in english and mandarin." International conference on machine learning. PMLR, 2016.

---

> > ### Comment · Reviewer_vUjU · 2021-11-23
> > **Keeping the score**
> >
> > The authors conducted additional experiments and it seems convincing to me.
> > It could have been more convincing if the authors have tried the proposed method on raw-waveform models but it's okay.
> > Thank you for your efforts.

---

### Official Review · Reviewer_rkdE · 2021-11-01

**Correctness:** 2
**Technical Novelty And Significance:** 3
**Empirical Novelty And Significance:** 2
**Recommendation:** 5
**Confidence:** 3

**Details Of Ethics Concerns:**

Generating adversarial noise sources which cannot be perceived by humans could potentially lead to several malignant applications.

**Main Review:**

Overall, I believe that the paper has great potential since the idea of energy dissipation is quite interesting but there are several concerns that need to be addressed. If the following concerns are addressed, I will consider to increase my score. Please see below my concerns and questions to the authors in descending order of importance (0 is the most important):

0. The main problem of this paper stems from the limited experimental setup that the authors have chosen to show the significance of their contributions. Specifically, the authors have only tested their algorithm for generating sequences with 100% WER and with a white-box ASR which was trained by the authors. It would be interesting to see if those adversarial examples are valid for pre-trained black box models even with not 100% WER. Moreoever, the 100% WER might be misleading since it would make much more sense to have a small change in a few words that would change the meaning than completely altering the whole target sentence. Furthermore, it might be the case that the other algorithms could alter a couple of words fairly easily but not all of them.

1. The authors provide only a very limited amount of adversarial examples in their supplementary material (which the authors also do not provide details if they are random or cherry-picked). Even from those limited examples I would admit that there are evident artifacts in PhaseFool’s adversarial examples. To the best of my hearing ability I would also say that the adversarial examples coming from the “Imperceptable” sound less artifact-y. I do not consider that the difference is so significant but I would like to hear more random examples to conclude.

2. There are several premises and claims in this paper which are at least ill-posed and sometimes wrong. For example, “Since phase is not relevant for speech recognition, the existing audio adversarial attacks neglect the influence of phase spectrogram.” is not true in cases where there are several speakers and a model (or even a human) uses the phase (aka inter-aural time-difference) between the two sources to better isolate and understand the words spoken by a speaker from a specific location. Another example, “Till now, we reveal the bi-directional essence of phase-oriented audio adversarial attack: it dissipates the critical energy for the classification model and constructs the adversarial perturbations in the meantime.” How is the latter statement shown? To the best of my knowledge, there is no “known” frequency region critical energy for classification for all ASR models.

3. I think that calling the inference/convergence times “efficiency (s)” is rather misleading. Moreover, the authors should report the number of trainable parameters, as well as the actual memory requirements in order to conclude that their algorithm is indeed more efficient and also discuss ways if their algorithm could be somehow more efficiently parallelized.

4. I am not fully convinced that an ASR model which does not use an STFT or a log-mel representation would also be vulnerable to an attack such as the one proposed in this paper. Specifically, with a small and simple adaptive front-end, one could possibly learn some real over-complete basis to be more robust against attacks like  PhaseFool. Real-basis have been successfully applied towards separation and enhancement problems but also show that can effectively represent speech with even a single layer [1, 2, 3, 4]. Could the authors provide ablation studies by replacing the fixed STFT layers with adaptive convolutions and still show that their findings still hold?

5. Based on the aforementioned point, the vulnerability of ASR models to phase perturbations or echoes is known for many years now [5] which partly explains the efficiency of PhaseFool that can easily drive the ASR model towards a mis-prediction while also not altering the structure of the sound significantly. Can PhaseFool be applied in reverberant scenarios where there are also other phase artifacts and still cannot be perceived by the human ear? Optionally, it would be wise to assume that there is at least some naive Wiener or minimum variance distortionless response (MVDR) filter and see how and if PhaseFool can trick those front-ends as well. There are other more sophisticated defences that the authors could also try, such as [6], to show that the proposed adversarial examples are not easily detectable by automated systems.

6. Table 1. should be WER 100% and the arrow next to WER should be facing up since  all of them are adversarial algorithms.

7. Could the authors specify what are the exact time-lengths of the input audio that the ~30seconds inference time corresponds to?

[1] Venkataramani S, Casebeer J, Smaragdis P. End-to-end source separation with adaptive front-ends. In2018 52nd Asilomar Conference on Signals, Systems, and Computers 2018 Oct 28 (pp. 684-688). IEEE.

[2] Luo, Y. and Mesgarani, N., 2019. Conv-tasnet: Surpassing ideal time–frequency magnitude masking for speech separation. IEEE/ACM transactions on audio, speech, and language processing, 27(8), pp.1256-1266.

[3] Tzinis, E., Venkataramani, S., Wang, Z., Subakan, C. and Smaragdis, P., 2020, May. Two-step sound source separation: Training on learned latent targets. In ICASSP 2020-2020 IEEE International Conference on Acoustics, Speech and Signal Processing (ICASSP) (pp. 31-35). IEEE.

[4] Mimilakis, S.I., Drossos, K. and Schuller, G., 2021, January. Unsupervised interpretable representation learning for singing voice separation. In 2020 28th European Signal Processing Conference (EUSIPCO) (pp. 1412-1416). IEEE.

[5] Xiong, F., Meyer, B.T., Moritz, N., Rehr, R., Anemüller, J., Gerkmann, T., Doclo, S. and Goetze, S., 2015. Front-end technologies for robust ASR in reverberant environments—spectral enhancement-based dereverberation and auditory modulation filterbank features. EURASIP Journal on Advances in Signal Processing, 2015(1), pp.1-18.

[6] Jayashankar, T., Roux, J.L., Moulin, P. (2020) Detecting Audio Attacks on ASR Systems with Dropout Uncertainty. Proc. Interspeech 2020, 4671-4675, doi: 10.21437/Interspeech.2020-1846.

===== POST - REBUTTAL =========
The authors have addressed some of my concerns, thus, I increase my score from 3 -> 5.

**Summary Of The Paper:**

This paper introduces a new way of constructing adversarial examples for automatic speech recognition using the phase of the short-time Fourier transformation. The authors conduct experiments and subjective evaluations to show how their adversarial examples compare to previous state-of-the-art algorithms. The experimental results show that the proposed phase-oriented audio adversarial samples can be produced in reduced time as well as they are less perceivable by humans.

**Summary Of The Review:**

Although the idea proposed in the paper is interesting and could have a great potential in the way that adversarial examples are generated for ASR systems. However, the current version of the paper displays only a limited experimental setup of the proposed algorithm that does not help to show the significance of the algorithm. I am willing to increase my score if my above concerns are addressed.

---

> ### Author Response · Authors · 2021-11-18
> **Response to Reviewer rkdE**
>
> First of all, thanks for your helpful comments!
>
> ### [About the limited experimental setup]
>
> We conduct the evaluations for the pretrained DeepSpeech 2 [1] ASR system. These results are shown in the later parts of our reply. Besides, we also evaluate the robustness of PhaseFool against several state-of-the-art defenses and environmental distortions (reverberations and white Gaussian noise) and the results are shown in the later parts of our reply. We have added these evaluations to the rebuttal version of the paper.
>
> *[Annotation] You can refer to Subsection 5.1 and Subsection 5.2 in our rebuttal version of the paper to see the detailed information.*
>
> | Method       | DeepSpeech 2 （Noisy or not） | Transformer（Noisy or not） |
> | ------------ | ----------------------------- | --------------------------- |
> | Clean        | 31%                           | 31%                         |
> | C&W's Attack | 75%                           | 77%                         |
> | IPC Attack   | 59%                           | 68%                         |
> | Qin's Attack | 49%                           | 45%                         |
> | PhaseFool    | 43%                           | 46%                         |
>
> | Method       | DeepSpeech 2 （Convergence Times (seconds)） | Transformer（Convergence Times (seconds)） |
> | ------------ | -------------------------------------------- | ------------------------------------------ |
> | C&W's Attack | 1031.16                                      | 211.87                                     |
> | IPC Attack   | 871.72                                       | 158.75                                     |
> | Qin's Attack | 1310.86                                      | 305.91                                     |
> | PhaseFool    | 197.85                                       | 33.10                                      |
>
> ### [About the calculation of WER]
>
> We choose the **targeted transcription** as the label for calculating the WER in our experiments following the previous works [2,3,4,5], so the best WER should be 0% and the arrow next to WER should be facing down like [5]. For example, if the targeted transcription only has a small change in a few words, this calculation is correct and meaningful. And we have defined these in Subsection 4 Evaluation Metrics, line 3 in the paper. I know that better WER can be higher for adversarial algorithms since they choose the original transcription as the label for calculating the WER (for example, a recent work in audio adversarial attack [6]). But we think the calculation method of WER in most of the previous works [2,3,4,5] is more meaningful. The detailed calculation method of WER is described in Subsection 4 Evaluation Metrics.
>
> ### [About evaluation with pre-trained black box models]
>
> Our work aims at finding new perspectives for audio adversarial attacks. Since most of the previous works [2,3,4,5] of the white-box audio adversarial attacks do not conduct transferability analysis and their methods has been widely applied to the black-box attacks [7] (some works even successfully deceive the online commercial ASR systems [8,9]), here we do not conduct the adversarial transferability analysis and leave it for our future work.
>
> ### [About the limited amount of adversarial examples in the supplementary material]
>
> We have attached more adversarial examples and their labels in the experiments of different attacks against the DeepSpeech 2 +CTC system in the supplementary material (in the folder named "more_examples_deepspeech2"). These examples are randomly selected from the 100 samples generated in the imperceptibility experiments. Furthermore, we have also attached the examples in the robustness evaluation in the reverberant and noisy environments.
>
> ### [About the ill-posed and wrong claims in the paper]
>
> Thanks for your suggestions!
>
> First of all, we are sorry for our error claim "Since phase is not relevant for speech recognition". This claim is referred from one important work [10] ( page 6, Section 3, E.3, line 2) in audio adversarial attacks, and we think this claim is wrong nowadays.
>
> We can not change the texts shown in the "Abstract" in the OpenReview system. But we have changed "Since phase is not relevant for speech recognition, the existing audio adversarial attacks neglect the influence of phase spectrogram." to "However, the existing audio adversarial attacks neglect the influence of phase spectrogram". And we have changed “Till now, we reveal the bi-directional essence of phase-oriented audio adversarial attack: it dissipates the critical energy for the classification model and constructs the adversarial perturbations in the meantime” to " Till now,  we  reveal  the  essence  of  phase-oriented  audio  adversarial  attack:  it  slightly  dissipates  the energy structure in the spectrogram and constructs the adversarial examples in the meantime." These changes are shown in red in our revised version of the paper.

---

> > ### Author Response · Authors · 2021-11-18
> > **Due to the limited number of characters, we show the rest of our reply here.**
> >
> > ### [About the “efficiency (s)” and details like trainable parameters]
> >
> > We call the inference/convergence times “efficiency (s)” following the previous work [5]. We have changed "efficiency (s)" to "convergence times (s)" in the rebuttal version of the paper.
> >
> > The trainable parameters of PhaseFool and baseline attacks are only equal to the length of the original audio. The computational complexity and memory requirements of an audio adversarial attack mainly depend on iteration steps and the attacked model f(·).  Since our PhaseFool is a one-stage attack with simple constraints to make the generated examples imperceptible, it only requires 700 steps to obtain a successful adversarial example against the DeepSpeech 2 system. So the generation speed of PhaseFool is much faster than the baseline attacks.
> >
> > ### [About the effectiveness against an ASR model which does not use an STFT or a log-mel representation]
> >
> > The previous works of audio adversarial attacks only investigate their effectiveness against ASR systems that use STFT, log-mel, or filterbank-related representations. And their methods have been widely applied to deceive the online commercial ASR systems [8,9]. Since replacing the fixed STFT layers with adaptive convolutions is not very common in audio adversarial attacks, we do not provide the results of this ablation study. Instead, we evaluate the robustness of PhaseFool against several sophisticated defenses. The results are shown in the later parts of our reply and are detailed in Subsection 5.3 in the rebuttal version of the paper.

---

> > > ### Author Response · Authors · 2021-11-18
> > > **Due to the limited number of characters, we show the rest of our reply here.**
> > >
> > > ### [About applying PhaseFool in reverberant scenarios or detecting PhaseFool with some sophisticated defense]
> > >
> > > We evaluate the robustness of PhaseFool against environmental distortions (reverberations and white Gaussian noise) and several state-of-the-art defenses (including [11]). The results are shown in the following tables. We have added these evaluations to the rebuttal version of the paper.
> > >
> > > *（Here we only implement the attacks against DeepSpeech 2 + CTC system in the robustness evaluations. You can refer to Subsection 5.3 in our rebuttal version of the paper to see the detailed information.）*
> > >
> > > *[Annotation] Here we list the AUC scores of these defenses.* The detailed setup and explanation can be found in Subsection 5.3 and Appendix.
> > >
> > > | Method       | Temporal Dependency Defense | Dropout Defense | Down Sampling Defense | Noise Defense |
> > > | ------------ | --------------------------- | --------------- | --------------------- | ------------- |
> > > | C&W's Attack | 57.78%                      | 63.75%          | 67.05%                | 63.70%        |
> > > | IPC Attack   | 52.67%                      | 71.62%          | 68.95%                | 74.42%        |
> > > | Qin's Attack | 58.22%                      | 61.20%          | 66.74%                | 82.64%        |
> > > | PhaseFool    | 57.02%                      | 64.09%          | 68.60%                | 62.81%        |
> > >
> > > *[Annotation] Here we list the results of robustness evaluation in the reverberant environment.*
> > >
> > > | Method       | Success Rate | WER    | Noisy or not |
> > > | ------------ | ------------ | ------ | ------------ |
> > > | C&W's Attack | 47%          | 22.69% | 100%         |
> > > | IPC Attack   | 21%          | 37.07% | 97%          |
> > > | Qin's Attack | 43%          | 26.44% | 85%          |
> > > | PhaseFool    | 38%          | 29.62% | 61%          |
> > >
> > > *[Annotation] Here we list the results of robustness evaluation in the noisy environment.*
> > >
> > > | Method       | Success Rate | WER    | Noisy or not |
> > > | ------------ | ------------ | ------ | ------------ |
> > > | C&W's Attack | 55%          | 15.38% | 88%          |
> > > | IPC Attack   | 37%          | 27.94% | 73%          |
> > > | Qin's Attack | 19%          | 43.45% | 51%          |
> > > | PhaseFool    | 44%          | 16.34% | 48%          |
> > >
> > >
> > >
> > > ### [About the Exact time-lengths of the input audio]
> > >
> > > The lengths of the input audio range from 3 seconds to 10 seconds (the average length is 5.18 seconds). We detailed this information in the Appendix in the rebuttal version of the paper.
> > >
> > > [*] We also include other details and explanations of the experiments in the Appendix.
> > >
> > >
> > >
> > > [1] Amodei, Dario, et al. "Deep speech 2: End-to-end speech recognition in english and mandarin." International conference on machine learning. PMLR, 2016.
> > >
> > > [2] Carlini, Nicholas, and David Wagner. "Audio adversarial examples: Targeted attacks on speech-to-text." 2018 IEEE Security and Privacy Workshops (SPW). IEEE, 2018.
> > >
> > > [3] Qin, Yao, et al. "Imperceptible, robust, and targeted adversarial examples for automatic speech recognition." International conference on machine learning. PMLR, 2019.
> > >
> > > [4] Yakura, Hiromu, and Jun Sakuma. "Robust audio adversarial example for a physical attack." IJCAI. 2019.
> > >
> > > [5] Liu, Xiaolei, et al. "Weighted-sampling audio adversarial example attack." *Proceedings of the AAAI Conference on Artificial Intelligence*. Vol. 34. No. 04. 2020.
> > >
> > > [6] Esmaeilpour, Mohammad, Patrick Cardinal, and Alessandro Lameiras Koerich. "Towards robust speech-to-text adversarial attack." *arXiv preprint arXiv:2103.08095* (2021).
> > >
> > > [7] Taori, Rohan, et al. "Targeted adversarial examples for black box audio systems." 2019 IEEE Security and Privacy Workshops (SPW). IEEE, 2019.
> > >
> > > [8] Yuan, Xuejing, et al. "Commandersong: A systematic approach for practical adversarial voice recognition." 27th {USENIX} Security Symposium ({USENIX} Security 18). 2018.
> > >
> > > [9] Chen, Yuxuan, et al. "Devil’s whisper: A general approach for physical adversarial attacks against commercial black-box speech recognition devices." 29th {USENIX} Security Symposium ({USENIX} Security 20). 2020.
> > >
> > > [10] Schönherr, Lea, et al. "Adversarial attacks against automatic speech recognition systems via psychoacoustic hiding." NDSS. 2019.
> > >
> > > [11] Jayashankar, T., Roux, J.L., Moulin, P. (2020) Detecting Audio Attacks on ASR Systems with Dropout Uncertainty. Proc. Interspeech 2020, 4671-4675, doi: 10.21437/Interspeech.2020-1846.

---

> > > > ### Comment · Reviewer_rkdE · 2021-11-22
> > > > **Response to the Authors**
> > > >
> > > > First of all, I would like to thank the authors for their detailed response to my comments as well as their effort to address all of the reviewers' concerns and conduct more experiments. Although most of my concerns have been addressed there are still some problems with the current version of the paper/evaluation.
> > > >
> > > > > Our work aims at finding new perspectives for audio adversarial attacks. Since most of the previous works [2,3,4,5] of the white-box audio adversarial attacks do not conduct transferability analysis and their methods has been widely applied to the black-box attacks [7] (some works even successfully deceive the online commercial ASR systems [8,9]), here we do not conduct the adversarial transferability analysis and leave it for our future work.
> > > >
> > > > I do understand that other works have not been asked to perform such experiments but since there are multiple work for white-box approaches in the literature, I find it rather hard to recommend one more work which does not have a set of experiments working with real world scenarios (assuming that one has full access to the architecture and the weights is extremely restrictive and very far from real-world scenarios).
> > > >
> > > > > We evaluate the robustness of PhaseFool against environmental distortions (reverberations and white Gaussian noise)...
> > > >
> > > > What is the error imposed to the WER of the ASR model from pure reverberation and how this compares to your attack? Why not true ambient noise sources? I think some clarification in this part would also be needed here.
> > > >
> > > > > The previous works of audio adversarial attacks only investigate their effectiveness against ASR systems that use STFT, log-mel, or filterbank-related representations. And their methods have been widely applied to deceive the online commercial ASR systems [8,9]. Since replacing the fixed STFT layers with adaptive convolutions is not very common in audio adversarial attacks, we do not provide the results of this ablation study. Instead, we evaluate the robustness of PhaseFool against several sophisticated defenses.
> > > >
> > > > First of all, the filterbanks could be learnable (thus, learnable convolutional basis) and then be fixed. The proposed attack might have a hard time bypassing real basis (which model phase shifts in a different way) and this is the reason I propose this type of front-ends which are much more sophisticated as a defense from many of the evaluated defences (e.g. "downsampling" or "noise").
> > > >
> > > > > Here we list the AUC scores of these defenses. The detailed setup and explanation can be found in Subsection 5.3 and Appendix.
> > > >
> > > > Seems like all the attacks (including the proposed one) perform similarly compared to some naive and attack agnostic defences (least of my concerns about the paper).
> > > >
> > > > Moreover, I have mentioned in my first comments that only using a targeted atttack with 0% WER could also be misleading, IMHO 100% WER only tells a part of the story about an attacking method for multiple reasons that I have also explained in detail in my initial review. I think the paper would be more complete if there was an analysis of the trade off between computation and true WER of the ASR system.
> > > >
> > > > The authors could also change the metric name "Noisy or not" to "Classified as Noisy" as it is much more descriptive and easy to understand.
> > > >
> > > > Overall, I think the authors have done a great job during the rebuttal period but there are still significant work to be done in order to truly unfold the potential of this paper. I will increase my score to a 5 because of the current version of the paper has adressed several of my concerns.

---

> > > > > ### Author Response · Authors · 2021-11-29
> > > > > **Response to Reviewer rkdE**
> > > > >
> > > > > ### [About experiments of  transferability analysis]
> > > > >
> > > > > Attacking a black-box ASR system is harder and more complicated than attacking the black-box image recognition system [1]. For example, the ImageNet-1K dataset only has 1K classes and the adversarial examples generated by the white-box recognition models usually have relatively high transferability against the black-box models. However, the targeted audio adversarial examples should be designed for a time series of speech data. For the vocabulary size of 32, there are 32 * 32 * 32 * 32 * 32 possible results for a simple target phase with the length of 5 characters (for example, "Hello"). Thus it is hard to implement even a small change in a few words that would change the meaning. Additionally, the feature extraction, acoustic model, and language model usage make black-box audio adversarial examples even harder.
> > > > >
> > > > > We have tried to examine whether the generated white-box adversarial examples are valid for the pre-trained black-box Conformer model in the [ESPnet github repository](https://github.com/espnet/espnet). However, for all of the attacks in our evaluations, the WERs are around 90% ~ 95% and the CERs are around 65% ~ 70%. We list three examples in the experiments of adversarial transferability below.
> > > > >
> > > > > | The transcriptions of adversarial examples            | The target transcriptions                              |
> > > > > | ----------------------------------------------------- | ------------------------------------------------------ |
> > > > > | He wiv dptple rh rrrose sscctly that ll mhould te     | He was unable to decide exactly what it should be      |
> > > > > | I r f fot giitg ty tlf aommom o raaye  a fhepg ff fhe | I did not wrong myself so but i placed a wrong on thee |
> > > > > | I is a meey taid t                                    | It is a duty said i                                    |
> > > > >
> > > > > To make PhaseFool and the baseline attacks more effective against the black-box ASR systems in real-world scenarios, we should generate hundreds of examples and carefully pick the successful black-box examples following [5]. However, due to the limited time, we haven't finished our work yet, so we regard it as our future work.
> > > > >
> > > > >
> > > > >
> > > > > ### [About the WERs of the ASR models from pure reverberation and how this compares to our attack]
> > > > >
> > > > > The WERs of the ASR models from pure reverberation and pure white Gaussian noise are listed in the following table. The WERs in both environments are relatively better than our attack, but the success rate (or sentence-level accuracy) is worse than our attack. We think the reason is that the reverberation and noise affect all clean speech clips similarly. However, some of the adversarial examples are more robust towards the reverberation and noise while the others are not so robust. We will add more details and clarification in this part to the new version of the paper.
> > > > >
> > > > > | ASR systems  | Success Rate (Reverberation) | WER (Reverberation) | Success Rate  (Noise) | WER (Noise) |
> > > > > | ------------ | :--------------------------: | :-----------------: | :-------------------: | :---------: |
> > > > > | DeepSpeech 2 |             22%              |       17.40%        |          25%          |   12.33%    |
> > > > >
> > > > > *[Annotation] Here we list the results of PhaseFool in the robustness evaluation.*
> > > > >
> > > > > | Method    | Success Rate (Reverberation) | WER (Reverberation) | Success Rate (Noise) | WER (Noise) |
> > > > > | --------- | ---------------------------- | ------------------- | -------------------- | ----------- |
> > > > > | PhaseFool | 38%                          | 29.62%              | 44%                  | 16.34%      |
> > > > >
> > > > >
> > > > >
> > > > > ### [About why not true ambient noise sources?]
> > > > >
> > > > > As illustrated in [2], the audio adversarial examples in the physical world always encounter the reverberation and white Gaussian noise caused by playback or recording. To examine the robustness of our adversarial examples, we follow the experimental setup of [2,3] and choose the reverberation and white Gaussian noise. We think the experiments with true ambient noise sources are also an important part that should be investigated. We will add more evaluations and clarifications in this part to the new version of the paper.
> > > > >
> > > > >
> > > > > ### [About the experiments of bypassing the learnable convolutional basis]
> > > > >
> > > > > We agree that your concerns are right and thanks for your helpful suggestions.  We are now trying to replace the fixed STFT layers with adaptive convolutions for the DeepSpeech 2 system. Since the training of the ASR system is highly time-consuming, we have not finished our work yet. We will add these evaluations to the new version of the paper.

---

> > > > > > ### Author Response · Authors · 2021-11-29
> > > > > > **Due to the limited number of characters, we show the rest of our reply here.**
> > > > > >
> > > > > >
> > > > > >
> > > > > > ### [About the similar results of various defenses]
> > > > > >
> > > > > > Yes, the noise defense is naive and attack-agnostic. However, the boundary of noise is -0.02 to +0.02 and the values of the adversarial perturbations are usually ranged between -0.005 to +0.005. Thus, the adversarial perturbations will be greatly influenced by the introduced noise and meanwhile, the original speech is distorted slightly. As for the downsample defense, the perturbations introduced to the waveform in a frame are not distributed independently since the unit of speech recognition model is the frame. The downsample and upsample process will destroy the structure of the perturbations inside a frame. Thus the downsample defense is also strong enough. So the results of these naive defenses are similar to the two sophisticated defenses [4,5].
> > > > > >
> > > > > >
> > > > > >
> > > > > > ### [About the calculation of WER]
> > > > > >
> > > > > > Yes, we agree that only using a targeted attack with 0% WER could also be misleading. We will add more clarifications about the metrics in evaluations to the new version of the paper.
> > > > > >
> > > > > >
> > > > > > ### [About the analysis of the trade-off between computation and true WERs of the ASR systems.]
> > > > > >
> > > > > > The true WERs of the ASR systems are listed in the following table. We will add the detailed analysis of the trade-off between computation (like the computation speed with different lengths of inputs, MACs, and memory usage ...) and true WERs of the ASR systems to the new version of the paper.
> > > > > >
> > > > > > | ASR systems          | WER (LibriSpeech test-clean) | WER (selected 100 samples) |
> > > > > > | -------------------- | :--------------------------: | :------------------------: |
> > > > > > | DeepSpeech 2         |            7.06%             |           9.46%            |
> > > > > > | Low-rank Transformer |            7.35%             |           8.27%            |
> > > > > >
> > > > > >
> > > > > > ### [About changing the metric name "Noisy or not"]
> > > > > >
> > > > > > Thanks for your advice! We will change *"Noisy or not"* to *"Classified as Noisy"* in the new version of the paper.
> > > > > >
> > > > > > Finally, thanks for your helpful comments!
> > > > > >
> > > > > > [1] Chen, Yuxuan, et al. "Devil’s whisper: A general approach for physical adversarial attacks against commercial black-box speech recognition devices." 29th {USENIX} Security Symposium ({USENIX} Security 20). 2020.
> > > > > >
> > > > > > [2] Yakura, Hiromu, and Jun Sakuma. "Robust audio adversarial example for a physical attack." IJCAI. 2019.
> > > > > >
> > > > > > [3] Qin, Yao, et al. "Imperceptible, robust, and targeted adversarial examples for automatic speech recognition." International conference on machine learning. PMLR, 2019.
> > > > > >
> > > > > > [4] Jayashankar, T., Roux, J.L., Moulin, P. (2020) Detecting Audio Attacks on ASR Systems with Dropout Uncertainty. Proc. Interspeech 2020, 4671-4675, doi: 10.21437/Interspeech.2020-1846.
> > > > > >
> > > > > > [5] Yang, Zhuolin, et al. "Characterizing audio adversarial examples using temporal dependency." ICLR. 2019.

---

### Official Review · Reviewer_sPgr · 2021-11-04

**Correctness:** 3
**Technical Novelty And Significance:** 3
**Empirical Novelty And Significance:** 2
**Recommendation:** 5
**Confidence:** 4

**Main Review:**

This paper provides an interesting perspective to generate stealthy adversarial audios. The paper is easy to follow and the proposed method is well-motivated.
However, the evaluation of the proposed attack in the paper is a bit weak. For instance, the paper didn't evaluate the proposed attack against different adversarial audio defenses to evaluate the resiliency of the attack.
It would be helpful if the authors can test the generated adversarial audios against different ASR systems.
Further, it would be important and interesting to test the adversarial transferability and other properties of the generated adversarial audios.

**Summary Of The Paper:**

This paper proposes a phase-oriented algorithm PhaseFool to efficiently construct imperceptible audio adversarial attacks with energy dissipation.  The authors leverage the spectrogram consistency of STFT to adversarially transfer phase perturbations to the adjacent frames and dissipate the energy that is crucial for ASR systems. Empirical evaluations show that the attack effectiveness of the proposed attack is high.

**Summary Of The Review:**

Overall the paper is well-organized and the proposed method is interesting.
If the authors can add necessary evaluation, such as the attack effectiveness comparison with other attacks against the state of the art defenses; the adversarial transferability analysis, and the adversarial evaluation against different ASR systems, I would like the increase my score.

---

> ### Author Response · Authors · 2021-11-18
> **Response to Reviewer sPgr**
>
> Thanks for your comments!
>
> ### [About the attack effectiveness comparison against the state of the art defenses]
>
> We conduct further studies to compare PhaseFool with other attacks against the state-of-the-art defenses (including the dropout defense [1] mentioned by reviewer rkdE). Here we only implement the attacks against DeepSpeech 2 + CTC system [2]. The results are shown in the following tables. We have added these evaluations to Subsection 5.3 of the rebuttal version of the paper.
>
> *[Annotation] Here we list the AUC scores of the WER in these defenses. You can refer to the following sections in our rebuttal version of the paper to see the detailed information.*
>
> | Method       | Temporal Dependency Defense | Dropout Defense | Down Sampling Defense | Noise Defense |
> | ------------ | --------------------------- | --------------- | --------------------- | ------------- |
> | C&W's Attack | 57.78%                      | 63.75%          | 67.05%                | 63.70%        |
> | IPC Attack   | 52.67%                      | 71.62%          | 68.95%                | 74.42%        |
> | Qin's Attack | 58.22%                      | 61.20%          | 66.74%                | 82.64%        |
> | PhaseFool    | 57.02%                      | 64.09%          | 68.60%                | 62.81%        |
>
> ### [About the adversarial evaluation against different ASR systems]
>
> We conduct the evaluations for the pretrained DeepSpeech 2 [2] ASR system with CTC loss. The results are shown in the following tables.
>
> *[Annotation] You can refer to Subsection 5.1 and Subsection 5.2 in our rebuttal version of the paper to see the detailed information.*
>
> | Method       | DeepSpeech 2 （Noisy or not） | Transformer（Noisy or not） |
> | ------------ | ----------------------------- | --------------------------- |
> | Clean        | 31%                           | 31%                         |
> | C&W's Attack | 75%                           | 77%                         |
> | IPC Attack   | 59%                           | 68%                         |
> | Qin's Attack | 49%                           | 45%                         |
> | PhaseFool    | 43%                           | 46%                         |
>
> | Method       | DeepSpeech 2 （Convergence Times (seconds)） | Transformer（Convergence Times (seconds)） |
> | ------------ | -------------------------------------------- | ------------------------------------------ |
> | C&W's Attack | 1031.16                                      | 211.87                                     |
> | IPC Attack   | 871.72                                       | 158.75                                     |
> | Qin's Attack | 1310.86                                      | 305.91                                     |
> | PhaseFool    | 197.85                                       | 33.10                                      |
>
>
>
> ### [About adversarial transferability analysis]
>
> Our work aims at finding new perspectives for audio adversarial attacks. Since most of the previous works [3,4,5,6] of the white-box audio adversarial attacks do not conduct transferability analysis and their methods has been widely applied to the black-box attacks [7] (some works even successfully deceive the online commercial ASR systems [8,9]), here we do not conduct the adversarial transferability analysis and leave it for our future work.

---

> > ### Author Response · Authors · 2021-11-18
> > **Due to the limited number of characters, we show the rest of our reply here.**
> >
> >
> > ### [About the weak evaluation of the proposed attack]
> >
> > Except for the above experiments, we also conduct the evaluations of attacks against environmental distortions (reverberations and white Gaussian noise). The results are shown in the following tables. We have added these evaluations to the rebuttal version of the paper and shown them in blue.
> >
> > *[Annotation] Here we list the results of robustness evaluation in the reverberant environment.*
> >
> > | Method       | Success Rate | WER    | Noisy or not |
> > | ------------ | ------------ | ------ | ------------ |
> > | C&W's Attack | 47%          | 22.69% | 100%         |
> > | IPC Attack   | 21%          | 37.07% | 97%          |
> > | Qin's Attack | 43%          | 26.44% | 85%          |
> > | PhaseFool    | 38%          | 29.62% | 61%          |
> >
> > *[Annotation] Here we list the results of robustness evaluation in the noisy environment.*
> >
> > | Method       | Success Rate | WER    | Noisy or not |
> > | ------------ | ------------ | ------ | ------------ |
> > | C&W's Attack | 55%          | 15.38% | 88%          |
> > | IPC Attack   | 37%          | 27.94% | 73%          |
> > | Qin's Attack | 19%          | 43.45% | 51%          |
> > | PhaseFool    | 44%          | 16.34% | 48%          |
> >
> >
> >
> > [1] Jayashankar, T., Roux, J.L., Moulin, P. (2020) Detecting Audio Attacks on ASR Systems with Dropout Uncertainty. Proc. Interspeech 2020, 4671-4675, doi: 10.21437/Interspeech.2020-1846.
> >
> > [2] Amodei, Dario, et al. "Deep speech 2: End-to-end speech recognition in english and mandarin." International conference on machine learning. PMLR, 2016.
> >
> > [3] Carlini, Nicholas, and David Wagner. "Audio adversarial examples: Targeted attacks on speech-to-text." 2018 IEEE Security and Privacy Workshops (SPW). IEEE, 2018.
> >
> > [4] Qin, Yao, et al. "Imperceptible, robust, and targeted adversarial examples for automatic speech recognition." International conference on machine learning. PMLR, 2019.
> >
> > [5] Yakura, Hiromu, and Jun Sakuma. "Robust audio adversarial example for a physical attack." IJCAI. 2019.
> >
> > [6] Zhang, Hongting, et al. "Generating Robust Audio Adversarial Examples with Temporal Dependency." IJCAI. 2020.
> >
> > [7] Taori, Rohan, et al. "Targeted adversarial examples for black box audio systems." 2019 IEEE Security and Privacy Workshops (SPW). IEEE, 2019.
> >
> > [8] Yuan, Xuejing, et al. "Commandersong: A systematic approach for practical adversarial voice recognition." 27th {USENIX} Security Symposium ({USENIX} Security 18). 2018.
> >
> > [9] Chen, Yuxuan, et al. "Devil’s whisper: A general approach for physical adversarial attacks against commercial black-box speech recognition devices." 29th {USENIX} Security Symposium ({USENIX} Security 20). 2020.

---

> ### Author Response · Authors · 2021-11-29
> **Response to Reviewer sPgr**
>
> Dear Reviewer sPgr,
>
> Thanks for your helpful comments! We have made responses to you and please take a look and see whether we address your concerns.

---

### Official Review · Reviewer_BtbA · 2021-11-10

**Correctness:** 3
**Technical Novelty And Significance:** 3
**Empirical Novelty And Significance:** 2
**Recommendation:** 5
**Confidence:** 4

**Main Review:**

The main strength of the paper is the novelty of the energy dissipation.

Main weaknesses include:
1. As pointed out by reviewer rkdE, the claim that serves as the motivation "Since phase is not relevant for speech recognition, the existing audio adversarial attacks neglect the influence of phase spectrogram." is not true. It is not shown in the paper what this claim is based on. I guess perhaps the authors thought traditional filterbank acoustic features for monaural ASR only keep magnitudes, which is certainly true. However, phase information is certainly important for ASR not only with binaural or microphone array data but also for monaural -- this is possibly why ASR systems with raw-waveform features can outperform those with filterbank features in many recent studies.
2. The type of ASR input features used is missing in the paper (perhaps log-mel spectrograms from Fig. 1?). I traced into the paper that provides the ASR model architecture (Winata et al., 2020a), but did not find any clear evidence for the feature type. This is critical here since filterbank-related features (including MFCCs) do not have explicit phase information. Perturbing phase information can reduce the magnitude of the spectrogram is certainly good evidence, but doesn't this contradict with the claim "Since phase is not relevant for speech recognition, the existing audio adversarial attacks neglect the influence of phase spectrogram."?
3. The ASR method used in this paper is not common. The effectiveness of such adversarial attacks is certainly related to the detailed model architecture and how it was trained. I would expect the method may not work for NN-HMMs, CTC, and stateless RNN-T.
4. It is also difficult for me to understand how to use such white-box adversarial attack techniques for ASR in practice. Industrial ASR systems are often trained on an enormously large dataset with many data augmentation techniques applied, which should be a lot more robust against energy dissipation. I understand the authors rely on (Chen et al., 2020) to convert a white-box model to a black-box model. However, the methods proposed in (Chen et al., 2020) are not convincible enough to me, since most of the industrial ASR systems studied in that paper are possibly NN-HMMs and CTC, which should be robust against adversarial attacks.
5. Although the adversarial attack is mostly a machine learning research topic, the novelty and the required knowledge are mostly signal processing. Perhaps ICASSP would be a more suitable place to publish this work than ICLR.
6. Contains many mistakes and typos, for instance:

(a) "Denote the ASR classification model as f : {X} → Y , with X ∈ [0, 1]t×c being the audio space."

I guess c is the number of bits related to the floating number precision, but it is not defined at all.

(b) The reference (Winata et al., 2020a) and (Winata et al., 2020b) are the same paper.

(c) In the equation, L_{net} means n, e, t are three variables. Should be L_{\text{net}} instead.

...


**Summary Of The Paper:**

This paper proposed a novel idea that generates imperceptible adversarial attacks for ASR by perturbing the phase information. Proof that phase perturbations reduce the magnitude of the spectrogram is provided. White-box-threat-model-based experimental results showed the proposed method can generate effective adversarial examples for an academic Transformer-based ASR system with better efficiency than the state-of-the-art counterpart methods.

**Summary Of The Review:**

In conclusion, the paper presents a novel idea to perform adversarial attacks for audio-based systems, but the experiment setup and results need improvement. The amount of work required is perhaps more than what could be finished based on the time constraint.

---

> ### Author Response · Authors · 2021-11-18
> **Response to Reviewer BtbA**
>
> Thanks for your comments!
>
> ### [About the error claims in the paper]
>
> First of all, we are sorry for our error claim "Since phase is not relevant for speech recognition, the existing audio adversarial attacks neglect the influence of phase spectrogram". This claim is referred from one important work [1] ( page 6, Section 3, E.3, line 2) in audio adversarial attacks, and we think this claim is wrong nowadays. We can not change the texts shown in the "Abstract" in the OpenReview system. But we have changed "Since phase is not relevant for speech recognition, the existing audio adversarial attacks neglect the influence of phase spectrogram." to "However, the existing audio adversarial attacks neglect the influence of phase spectrogram." and show it in red in our revised version of the paper.
>
> ### [About the missing type of ASR input features used in the paper]
>
> Thanks for your concerns. The type of ASR input features for the low-rank transformer systems is log-mel spectrograms. We have added this information to the rebuttal version of the paper.  Again, we are sorry for our error claim.
>
>
> ### [About PhaseFool's effectiveness against common ASR systems]
>
> We agree that the ASR method used in this paper is not common. To address the concern that our method may not work for common ASR systems, we conduct the evaluations for the pretrained DeepSpeech 2 [2] ASR system with CTC loss to prove that our attack is suitable for common ASR systems like CTC-based systems. The results are shown in the following tables. We can see that PhaseFool can successfully attack the ASR systems that adopt CTC loss. We have added this evaluation to the rebuttal version of the paper.
>
> *[Annotation] You can refer to Subsection 5.1 and Subsection 5.2 in our rebuttal version of the paper to see the detailed information.*
>
> | Method       | DeepSpeech 2 （Noisy or not） | Transformer（Noisy or not） |
> | ------------ | ----------------------------- | --------------------------- |
> | Clean        | 31%                           | 31%                         |
> | C&W's Attack | 75%                           | 77%                         |
> | IPC Attack   | 59%                           | 68%                         |
> | Qin's Attack | 49%                           | 45%                         |
> | PhaseFool    | 43%                           | 46%                         |
>
> | Method       | DeepSpeech 2 （Convergence Times (seconds)） | Transformer（Convergence Times (seconds)） |
> | ------------ | -------------------------------------------- | ------------------------------------------ |
> | C&W's Attack | 1031.16                                      | 211.87                                     |
> | IPC Attack   | 871.72                                       | 158.75                                     |
> | Qin's Attack | 1310.86                                      | 305.91                                     |
> | PhaseFool    | 197.85                                       | 33.10                                      |
>
> ### [About the meaning of white-box adversarial attack and the effectiveness of  [6] against industrial ASR systems in practice]
>
> White-box adversarial attack techniques like [1,3,4] usually serve as the foundations for robust over-the-air attacks [5] or black-box attacks [6,7] and can inspire a lot of future works. And we are sure that our white-box attack will bring new directions for future works in audio adversarial attacks.
>
> Besides, Chen et al. have provided their implementations for the methods proposed in [6]. We have run their code and successfully attacked the online commercial ASR systems listed in their paper.
>
> ### [About the suitable conference for our work]
>
> Since the idea of energy dissipation is novel, our work can greatly inspire other adversarial attacks or serve as the theoretical basis for future works in this area. Besides, there are many impressive works [3,8] of audio adversarial examples published in the conference of machine learning like ICLR and ICML. Their novelty and the required knowledge are also signal processing like frequency masking or temporal dependency and they have inspired and influenced a lot of works. So we think ICLR is a suitable place to publish our work.
>
> ### [About the mistakes and typos]
>
> We are sorry for our mistakes and typos. We have fixed these mistakes and typos and marked them red in the rebuttal version of the paper.

---

> > ### Author Response · Authors · 2021-11-18
> > **Due to the limited number of characters, we show the rest of our reply here.**
> >
> >
> > ### [About the weak experimental setup and results]
> >
> > We conduct the evaluations for the pretrained DeepSpeech 2 [2] ASR system. Besides, we also evaluate the robustness of PhaseFool against several state-of-the-art defenses (including the dropout defense mentioned by reviewer rkdE) and environmental distortions (including reverberations and white Gaussian noise). The results are shown in the following tables. We have added these evaluations to the rebuttal version of the paper.
> >
> > *（Here we implement the attacks against DeepSpeech 2 + CTC system in the robustness evaluations. You can refer to Subsection 5.3 in our rebuttal version of the paper to see the detailed information.）*
> >
> > *[Annotation] Here we list the AUC scores of these defenses against different attacks.*
> >
> > | Method       | Temporal Dependency Defense | Dropout Defense | Down Sampling Defense | Noise Defense |
> > | ------------ | --------------------------- | --------------- | --------------------- | ------------- |
> > | C&W's Attack | 57.78%                      | 63.75%          | 67.05%                | 63.70%        |
> > | IPC Attack   | 52.67%                      | 71.62%          | 68.95%                | 74.42%        |
> > | Qin's Attack | 58.22%                      | 61.20%          | 66.74%                | 82.64%        |
> > | PhaseFool    | 57.02%                      | 64.09%          | 68.60%                | 62.81%        |
> >
> > *[Annotation] Here we list the results of robustness evaluation in the reverberant environment.*
> >
> > | Method       | Success Rate | WER    | Noisy or not |
> > | ------------ | ------------ | ------ | ------------ |
> > | C&W's Attack | 47%          | 22.69% | 100%         |
> > | IPC Attack   | 21%          | 37.07% | 97%          |
> > | Qin's Attack | 43%          | 26.44% | 85%          |
> > | PhaseFool    | 38%          | 29.62% | 61%          |
> >
> > *[Annotation] Here we list the results of robustness evaluation in the noisy environment.*
> >
> > | Method       | Success Rate | WER    | Noisy or not |
> > | ------------ | ------------ | ------ | ------------ |
> > | C&W's Attack | 55%          | 15.38% | 88%          |
> > | IPC Attack   | 37%          | 27.94% | 73%          |
> > | Qin's Attack | 19%          | 43.45% | 51%          |
> > | PhaseFool    | 44%          | 16.34% | 48%          |
> >
> > [1] Schönherr, Lea, et al. "Adversarial attacks against automatic speech recognition systems via psychoacoustic hiding." NDSS. 2019.
> >
> > [2] Amodei, Dario, et al. "Deep speech 2: End-to-end speech recognition in english and mandarin." International conference on machine learning. PMLR, 2016.
> >
> > [3] Qin, Yao, et al. "Imperceptible, robust, and targeted adversarial examples for automatic speech recognition." International conference on machine learning. PMLR, 2019.
> >
> > [4] Carlini, Nicholas, and David Wagner. "Audio adversarial examples: Targeted attacks on speech-to-text." 2018 IEEE Security and Privacy Workshops (SPW). IEEE, 2018.
> >
> > [5] Yakura, Hiromu, and Jun Sakuma. "Robust audio adversarial example for a physical attack." IJCAI. 2019.
> >
> > [6] Chen, Yuxuan, et al. "Devil’s whisper: A general approach for physical adversarial attacks against commercial black-box speech recognition devices." 29th {USENIX} Security Symposium ({USENIX} Security 20). 2020.
> >
> > [7] Yuan, Xuejing, et al. "Commandersong: A systematic approach for practical adversarial voice recognition." 27th {USENIX} Security Symposium ({USENIX} Security 18). 2018.
> >
> > [8] Yang, Zhuolin, et al. "Characterizing audio adversarial examples using temporal dependency." ICLR. 2019.

---

> > > ### Comment · Reviewer_BtbA · 2021-11-24
> > > **Increasing my score from 3 to 5**
> > >
> > > I appreciate the efforts and the great job the authors have done during the rebuttal period and I am happy to increase my score from 3 to 5. I still have concerns about the settings. First, as far as I know DeepSpeech2 is highly cited but not widely used as research and commercial ASR systems. Second, the authors built their own models on LibriSpeech, but there is no baseline system WERs provided (without any attacks) which makes it impossible to find out the quality of the ASR systems. Other detailed settings include feature dimensions, frame length, and frame-shift are also missing.

---

> > > > ### Author Response · Authors · 2021-11-29
> > > > **Response to Reviewer BtbA**
> > > >
> > > > ### [About the reason we choose DeepSpeech 2]
> > > >
> > > > The DeepSpeech 2 system is widely used in the field of adversarial examples and the CNN+RNN+CTC architecture is also widely adopted in other ASR systems. So we conduct evaluations against the DeepSpeech 2 system to make our experiments more convincing. Thanks for your comments. We are now trying to conduct evaluations against the pretrained Conformer-based system (which is widely used in research and commercial scenarios) in the **[ESPnet github repository](https://github.com/espnet/espnet)** to make our experiments more convincing. We will add these evaluations to the new version of the paper.
> > > >
> > > > ### [About WERs of the baseline systems (without any attacks) ]
> > > >
> > > > As shown in Footnote 2 on page 6, we use the pretrained DeepSpeech 2 model adopted in the official github repository of **[adversarial-robustness-toolbox](https://github.com/Trusted-AI/adversarial-robustness-toolbox)** [1]. And we build our own low-rank Transformer model on LibriSpeech following the github repository **[end2end-asr-pytorch](https://github.com/gentaiscool/end2end-asr-pytorch)**. The WERs of the baseline ASR systems (without any attacks) are listed below. We also provide the WERs of the ASR model (without any attacks) in reverberant and noisy environments in the robustness evaluations. We will add these tables and more details to the new version of the paper.
> > > >
> > > > | ASR systems          | WER (LibriSpeech test-clean) | WER (selected 100 samples) |
> > > > | -------------------- | :--------------------------: | :------------------------: |
> > > > | DeepSpeech 2         |            7.06%             |           9.46%            |
> > > > | Low-rank Transformer |            7.35%             |           8.27%            |
> > > >
> > > > | ASR systems  | WER (Reverberation) | WER (Noise) |
> > > > | ------------ | :-----------------: | :---------: |
> > > > | DeepSpeech 2 |       17.40%        |   12.33%    |
> > > >
> > > > ### [About the missing detailed settings of the feature extraction]
> > > >
> > > > The detailed settings of the feature extraction for ASR systems in the paper are listed in the following table. And the n_fft, window_length, hop_length of PhaseFool attack are set as 1024, 1024, 512. We will add more details and clarifications to the new version of the paper.
> > > >
> > > > | ASR systems          | n_fft | window_length | hop_length |
> > > > | -------------------- | :---: | :-----------: | :--------: |
> > > > | DeepSpeech 2         |  320  |      320      |    160     |
> > > > | Low-rank Transformer |  320  |      320      |    160     |
> > > >
> > > >
> > > >
> > > > ### reference
> > > >
> > > > [1] Nicolae, Maria-Irina, et al. "Adversarial Robustness Toolbox v1.0.0." *arXiv preprint arXiv:1807.01069* (2018).

---

> > > > > ### Comment · Reviewer_BtbA · 2021-11-29
> > > > > **Thanks for the efforts & update**
> > > > >
> > > > > Thanks for the efforts & update. I will be happy to increase my scores further if your next version includes more comprehensive experiments with CTC, LF-MMI, RNN-T, and LAS systems (with both Conformer encoder and LSTM encoder to show the impact of energy dissipation to fully connected and CNN layers, and to different decoders). It's useful to make the baseline WERs similar to the best ones in each setting since the current DeepSpeech 2 and Low-rank Transformer Librispeech baseline WERs are not satisfactory.

---

> > > > > > ### Author Response · Authors · 2021-11-29
> > > > > > **Response to Reviewer BtbA**
> > > > > >
> > > > > > Thanks for your suggestions! We will add more comprehensive experiments with different systems and different encoder-decoder architectures in our next version of the paper and make the baseline WERs similar to the best ones in each setting!

---

### Comment · Area_Chair_vuqy · 2021-11-15
**Please address reviewer's comments**

Hi Authors,

Please address the reviewers's comments. Thanks!

---

> ### Author Response · Authors · 2021-11-16
> **About replies to reviewers**
>
> Thanks for your comment. We are trying to conduct more experiments following the reviewers' comments and concerns. Since the amount of work is heavy, we have not finished our work yet.

---

### Author Response · Authors · 2021-11-18
**Summary of the rebuttal revision**

We would like to thank the reviewers for their constructive reviews! We have revised the paper according to the reviewers' comments and provided detailed responses to each reviewer. A summary of the modification in the revised paper is as follows:

(1) We have fixed the error claims and mistakes in this paper and shown them in red. Besides, the added explanations are shown in blue.

(2) We have implemented our attack with the DeepSpeech 2 ASR system [1] with CTC loss to prove that our attack is suitable for common ASR systems like CTC-based systems.

(3) We have also conducted robustness evaluations against several defenses and environmental distortions (room reverberation and white Gaussian noise) to investigate the performance of our attack in simulated practical scenarios.

(4) More details like the setup of the robustness evaluations, the lengths of the selected samples, and convergence analysis are listed in the Appendix due to the limited page space.


[1] Amodei, Dario, et al. "Deep speech 2: End-to-end speech recognition in english and mandarin." International conference on machine learning. PMLR, 2016.

---

### Comment · Area_Chair_vuqy · 2021-11-24
**Please update your ratings if needed based on the authors' responses**

Dear Reviewers,

The authors have made detailed responses to all the reviews. Please take a look and see whether they address your concerns and update the ratings if necessary. Thanks for your help and expertise!

---

### Decision · Program_Chairs · 2022-01-20

**Decision:**

Reject

**Comment:**

This paper proposed a novel phase-oriented algorithm to efficiently construct imperceptible audio adversarial attacks. It leverages the spectrogram consistency of STFT to adversarially transfer phase perturbations to the adjacent frames and dissipate the energy that is crucial for ASR systems. Empirical evaluations show that the attack effectiveness of the proposed attack is high.

As agreed by the reviewers the method is very interesting, but the experimental justification is limited, lacking strong SOTA baseline ASR systems, different ASR model architectures, the adversarial transferability analysis etc. The author did add DeepSpeech 2 results to the initial version's Low-rank Transformer only results, which is still not convincing enough. The author also commented that they will "add more comprehensive experiments with different systems and …architectures in our next version of the paper" which is not in the current paper yet. Hence, resubmission with more experimental evaluations is recommended.

The decision is mainly due to the weak experimental justification.